# DNA2 and MSH2 cooperatively repair stabilized G4 and allow efficient telomere replication

Anthony Fernandez[1,10], Tingting Zhou [1,10], Yi Lei[1,10], Nian Liu[1,10], Steven Esworthy[1], Changxian Shen[1], Helen Liu[1,2], Jessica D. Hess [1], Hang Yuan[1], Guojun Shi[1], Mian Zhou[1], Lei Shen[1], Sufang Zhang [3], Settapong Kosiyatrakul[4], Vikas Gaur [5], Joshua A. Sommers[6], Nityanand Srivastava[7], Winfried Edelmann[7], Guo-Min Li [8], Robert M. Brosh Jr [6], Weihang Chai[5], Marietta Y. W. T. Lee[3], Dong Zhang [9], Carl Schildkraut[4], Li Zheng [1] ✉ & Binghui Shen [1] ✉

G-quadruplexes (G4s) are widely existing stable DNA secondary structures in mammalian cells. A long-standing hypothesis is that timely resolution of G4s is needed for efficient and faithful DNA replication. In vitro, G4s may be unwound by helicases or alternatively resolved via DNA2 nuclease mediated G4 cleavage. However, little is known about the biological significance and regulatory mechanism of the DNA2-mediated G4 removal pathway. Here, we report that DNA2 deficiency or its chemical inhibition leads to a significant accumulation of G4s and stalled replication forks at telomeres, which is demonstrated by a high-resolution technology: Single molecular analysis of replicating DNA (SMARD). We further identify that the DNA repair complex MutSα (MSH2-MSH6) binds G4s and stimulates G4 resolution via DNA2-mediated G4 excision. MSH2 deficiency, like DNA2 deficiency or inhibition, causes G4 accumulation and defective telomere replication. Meanwhile, G4-stabilizing environmental compounds block G4 unwinding by helicases but not G4 cleavage by DNA2. Consequently, G4 stabilizers impair telomere replication and cause telomere instabilities, especially in cells deficient in DNA2 or MSH2.

Efficient and faithful DNA replication is of paramount importance to cell survival and proper proliferation[1–6]. However, DNA replisomes frequently face challenges, including endogenous factors and/or environmental DNA-damaging agents[1,7–10]. These stress conditions cause DNA replication forks to stall, and if not stabilized or repaired, they collapse, resulting in mutations and chromosomal rearrangements, cell cycle arrest, or cell death. Repetitive DNA sequences such as micro-satellites or mini-satellites in centromeres, telomeres, and

[1]Department of Cancer Genetics and Epigenetics, Beckman Research Institute, City of Hope, Duarte, CA, USA. [2]Department of Ecology and Evolutionary Biology, College of Letters and Science, University of California at Los Angeles, Los Angeles, CA, USA. [3]Department of Biochemistry and Molecular Biology, New York Medical College, Valhalla, NY, USA. [4]Department of Cell Biology, Albert Einstein College of Medicine, Bronx, NY, USA. [5]Center for Genetic Diseases, Rosalind Franklin University of Medicine and Science, North Chicago, IL, USA. [6]Helicases and Genomic Integrity Section, Translational Gerontology Branch, National Institute on Aging, Bethesda, MD, USA. [7]Department of Cell Biology, Albert Einstein College of Medicine, Bronx, NY, USA. [8]Institute for Cancer Research, Chinese Institutes for Medical Research, Beijing, China. [9]Department of Biomedical Sciences, College of Osteopathic Medicine, New York Institute of Technology, Old Westbury, NY, USA. [10]These authors contributed equally: Anthony Fernandez, Tingting Zhou, Yi Lei, Nian Liu. ✉e-mail: lzheng@coh.org; bshen@coh.org

other loci may potentially form non-β-form DNA structures, which are "difficult-to-replicate" (DTR)[11–15]. They are major endogenous sources of replication stress. Of the non-B-form DNA structures, G-quadruplexes (G4s) can spontaneously form and are thermodynamically stable. The G-rich single-strand DNA (ssDNA) sequences adopt a conformation which consists of stacks of two or more G-quartets formed by four guanines via Hoogsteen base-pairing stabilized by a monovalent cation[16–19].

In mammalian cells, G4s are prevalent as an epigenetic structural and functional motif across the genome[20,21]. G4s, if not resolved prior to DNA replication, may frequently cause problems for DNA polymerases[22,23]. Different mechanisms exist in mammalian cells to resolve and/or clean up the G4s barrier for efficient DNA replication. DNA helicases such as FANCJ and BLM can unwind G4s[24,25]. In addition, the ssDNA binding protein RPA (RPA1, RPA2, RPA3) or CST (CTC1, STN1, TEN1) may possibly unfold G4 structures[26–28]. It is known that unfolding or unwinding G4s typically requires a ssDNA tail on which to load the RPA, CST, or G4 helicases[24,25,27]. These two mechanisms are crucial for resolving G4s under normal physiology. However, if cells are exposed to environmental chemical compounds (ECCs) that induce or stabilize G4s, helicase-driven G4s unwinding or RPA-mediated G4 melting are inhibited[24,25]. Several industrial ECCs, endogenous metabolites, or drugs, including TMPyP4, can bind to, stabilize, and inhibit the unfolding or unwinding of G4s[24,25], and newer technologies have permitted the discovery of a greater variety of G4 stabilizing compounds[29].

Previously, we showed that the bifunctional mammalian helicase/nuclease DNA2 cleaves G4s or other secondary structures and that such nucleolytic activity is important in promoting centromeric and telomeric DNA replication[30,31]. However, the biological significance of this DNA2-mediated G4 resolution pathway requires more investigation. The impact of G4-inducing/stabilizing ECCs on this pathway is unclear. Furthermore, because G4 structures are implicated as chromosomal structural elements and epigenetic motifs that regulate gene expression[20,21,32,33] DNA2-mediated G4s excision must be tightly controlled. Previous studies have reported many G4-binding proteins[34,35]. The function of these G4 binding proteins in the regulation of G4 resolution for DNA replication remains unknown.

In this study, we detect G4s in mammalian cells using a G4-specific antibody to determine that DNA2 deficiency or chemical inhibition leads to a significant accumulation of G4s. Furthermore, DNA2 deficiency potentiates G4 stabilizing compounds such as PIPER and TMPyP4 to increase G4 levels in cells. Using the single-molecule analysis of replicating DNA (SMARD) technique[36], we show that DNA2 deficiency and/or G4 stabilizing compounds result in replication defects in telomeres, which contain the most abundant G4-forming sequences in the genome[17]. We further identify that the mismatch repair protein MSH2 is a G4 binding protein in vitro and in cells. MSH2 is a key activator of DNA2 nuclease activity to cleave G4s in vitro and clean up G4s for DNA replication in mammalian cells. MSH2 is a member of the mismatch repair pathway for genome duplication fidelity. It forms a complex with MSH3 or MSH6 to recognize DNA mismatches or abnormal DNA structures[37]. We discover that MutSα (MSH2-MSH6) interacts with DNA2 and stimulates DNA2 to cleave G4s and facilitate Polδ-mediated DNA synthesis. Our result is consistent with the previous report that MutSα binds to G4 in vitro[38]. MSH2 deficiency, like DNA2 deficiency or inhibition, causes G4s accumulation and defective telomere replication. In addition, G4-inducing and -stabilizing compounds such as PIPER block G4 resolution by helicases and cause significant G4 accumulation, particularly in DNA2 and MSH2 mutant cells. Our studies suggest an important role of MSH2 in facilitating DNA2-mediated G4 removal, especially in the presence of G4-stabilizing compounds.

## Results

### DNA2 mediates G4 excision repair for efficient telomere DNA replication

We previously showed that purified recombinant DNA2 cleaved G4-bearing DNA substrates, mimicking the intramolecular G4 structures ahead of or within the DNA replication fork[31]. To determine if DNA2 is important for removing G4s in cells, we performed immuno-fluorescence (IF) co-staining of DNA2 and G4, using anti-DNA2 and an anti-G4 structure-specific antibody[22], and used an Airyscan Joint Deconvolution (jDCV) confocal microscope with the joint deconvolution algorithm to visualize DNA2 and G4 DNA foci at sub-diffraction-limited resolution. Prior to conducting the colocalization experiments, we validated the specificity of the G4 antibody by pre-incubating it with an excess of in vitro–folded bubble G4 substrate as a competitive inhibitor. This neutralization effectively abolished G4 staining, confirming that the antibody specifically recognizes G4 in cells (Supplementary Fig. 1).

To assess whether DNA2 and G4 signals are significantly associated in nuclei, we plotted identified DNA2 and G4 spots. We then measured all nearest distances between DNA2 and G4 spots and classified DNA2 spots that had a G4 signal within a 100-nm radius as co-localized. We rationalized that any spots found within the known resolution limit of the jDCV algorithm are significantly co-localized, which is far more stringent than colocalization analysis performed on standard confocal microscopes. We then normalized our measured co-localized foci by the densities of DNA2 and G4. We found a relative co-localization frequency that was ~25% greater than a random distribution of points in WT cells (Fig. 1a, b). This percentage is consistent with the fact that DNA2 quickly and efficiently resolves G4, and thus such interactions are not expected to be long-lived. To further demonstrate if DNA2 catalyzes G4 excision repair, we reconstituted G4 excision repair in an assay using purified recombinant DNA2 and polymerases and synthetic oligo based G4 DNA substrates (Fig. 1c). We first confirmed the formation of G-quadruplex with native PAGE electrophoresis in presence of KCl in contrast to LiCl (Supplementary Fig. 2). In the G4 excision repair assay, DNA2 cleavage of G4s results in a DNA gap. DNA polymerase Polδ or Polβ (Fig. 1c) then fills in the DNA gap with $^{32}$P-labeled dTTP and the other three deoxyribonucleotides, producing $^{32}$P-labeled gap-filled products and fully extended products. We found that DNA2 and DNA polymerases effectively repaired G4s, producing duplex DNA (extended products), but in the absence of DNA2, neither Polδ or Polβ produced any repair products (Fig. 1c). To test if DNA2 is important for G4 resolution in cells, we carried out G4 IF staining in wild-type (WT) mouse endothelial fibroblasts (MEFs) and *DNA2*⁺/⁻ MEFs, which we previously generated and used to study the function of DNA2 in G4 resolution and telomere replication. We also analyzed G4 levels in WT MEFs treated with the DNA2 inhibitor C5. We found that DNA2 haploinsufficiency or DNA2 inhibition by C5 greatly increased G4 levels (Fig. 1d, e). We also tested this G4 accumulation specifically at telomeres by co-staining cells with a telomere-specific FISH probe. We found that G4 accumulated significantly at telomeres. Further, we tested the WRN helicase inhibitor HRO761 and found that it too was able to increase G4 specifically at telomeres, as did the DNA2 inhibitor C5. The combination of these two inhibitors leads to a further increase of G4 positive telomeres (Supplementary Fig. 3), suggesting that in addition to the well-characterized helicase pathway of G4 resolution at telomeres, the DNA2-associated nuclease pathway is indispensable for telomere maintenance.

To further determine the impact of G4 accumulation on DNA replication in cells, we analyzed replication specifically at the telomere region, which consists of the repetitive DNA sequence TTAGGG and has a high propensity for forming G4 structures[39]. We reasoned

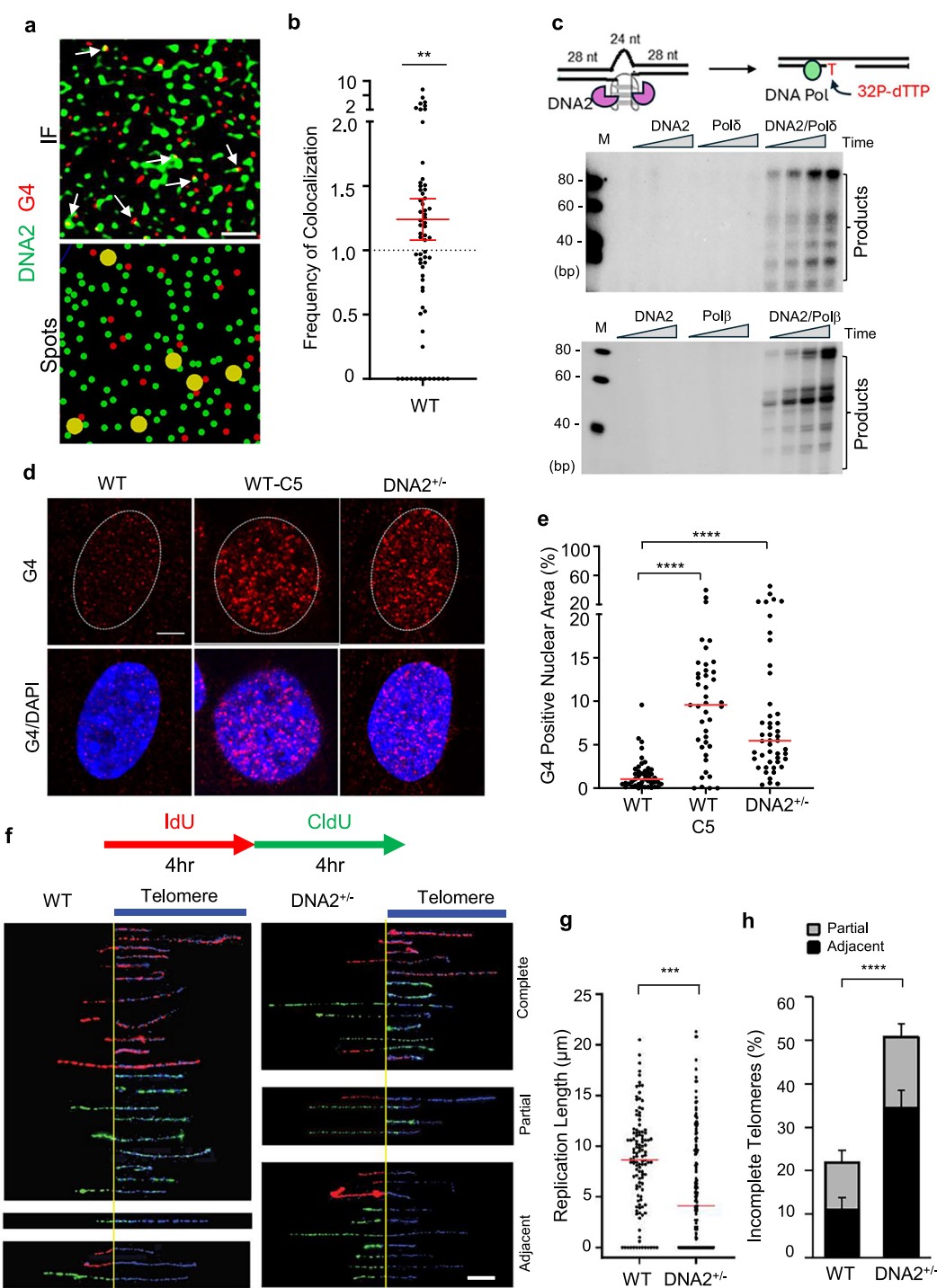

**Fig. 1 | DNA2 deficiency leads to G4 accumulation in mammalian cells. a** The top panel shows an AIRYSCAN joint deconvolution (jDCV) nuclear region with DNA2 in green and G4 in red. Scale bar = 1 μm. The bottom panel shows the identified spots of this region with DNA2 spots within 100 nm of a G4 spot in yellow; **b** Relative frequency of co-localization of DNA2 and G4. Red lines indicated mean ± SEM. P-value was calculated using a two-tailed paired t-test (n = 64 cells, p = 0.003); **c** Reconstituted G4 excision repair assay. Top panel: schematic diagram detailing DNA2 cleavage of G4 and gap formation. Polδ or Polβ then fills the gap using $^{32}$P-dTTP and other dNTPs, producing $^{32}$P-labeled products. Bottom panel: Representative denaturing PAGE image from 3 independent experiments showing G4 excision and repair; **d, e** G4 immunofluorescence staining in WT and DNA2$^{+/-}$ MEFs. **d** The representative AIRYSCAN confocal microscopy images of G4s. Scale bar = 5 μm, and **e** The quantification of G4 by condition. The red line indicates median. P-values were calculated using one-way ANOVA (p < 0.0001, n = 45, 42, 48 cells).;

**f–h** DNA replication at telomeres in WT and DNA2$^{+/-}$ MEFs; **f** Top: Scheme of the IdU (red) and CldU (green) pulse labeling. Bottom: SMARD microscopy images of replicated telomeres in WT and DNA2$^{+/-}$ MEF cells. SMARD fibers are arranged showing non-telomeric DNA at the left and telomeres on the right (indicated in blue). The top panels of SMARD fibers represent fully replicated telomeres, the middle partially replicated telomeres, and the bottom where replication halted immediately adjacent to telomeres. Scale = 5 μm; **g** Relative length of replicated telomeres (n = 118, 148 fibers). The red line indicates the median. P-value was calculated using two-sided Student's t-test (p = 0.0008); **h** Percentage of incompletely replicated telomeres and stalling adjacent to telomeres. Bars indicate mean and SD (n = 119, 148 fibers). P values were calculated using a two-sided Chi-squared test (p < 0.0001). *p < 0.05, **p < 0.01, ***p < 0.001, and ****p < 0.0001. Source data are provided as a Source data file. Created in BioRender. Zhou, T. (2025) https://BioRender.com/l90i3dw.

that this would be an effective way to assess the specific impact of G4 on replication in cells. Using the SMARD approach to analyze replication at the individual DNA fiber levels, we analyzed replication dynamics exclusively in telomere sequences, using thymidine analogs CldU and IdU to label nascent DNA and a telomere-specific probe to define the telomere regions (Fig. 1f). We found in the WT MEFs within 8 h that the median length of replicated telomeres was 8.7 μm, while in $DNA2^{+/-}$ MEFs it was 4.6 μm (Fig. 1g, h), exhibiting slower replication fork speeds in $DNA2^{+/-}$ MEFs. In addition, the frequency of incompletely replicated telomere (partial and adjacent) increases from 22% (11% partial and 11% adjacent) in WT MEFs to 50% (16% partial and 35% adjacent) in $DNA2^{+/-}$ MEFs (Fig. 1g, h). Consistently, inhibition of DNA2 by the DNA2 inhibitor C5[40] in the WT cells caused similar replication stalling at telomeres (Supplementary Fig. 4). This effect was noted to be exclusive to the telomere containing fibers, as no difference was found in replication rates in other, non-telomere containing fibers (Supplementary Fig. 5). While we have previously shown that DNA2 haploinsufficiency caused an increased rate of telomere defects, we also examined the effect of complete DNA2 loss in $DNA2^{-/-}$ embryonic stem (ES), where cells had significantly more telomere loss and sister telomere fusion than the WT ES cells (Supplementary Fig. 6).

## MutSα facilitates DNA2 binding to G4 structure and stimulates its G4 cleavage activities for telomere replication

We next explored the factors that contribute to DNA2-mediated repair of G4s. To this end, we expressed 3x-Flag-tagged DNA2 in 293T cells and carried out co-IP to pull down the 3x-Flag-tagged DNA2 (Fig. 2a) and its binding proteins. We used mass spectrometry to identify the DNA2-binding proteins. Our analysis revealed DNA replication and repair platform proteins such as RPA, RFC, and PCNA, which have previously been found in complex with DNA2[30,41,42] (Fig. 2b), as well as several previously unidentified binding partners, including MMR proteins MSH2 (the most prevalent protein in our screen) and MSH6; nucleotide excision repair (NER) proteins RAD23A and RAD23B; NHEJ proteins Ku70/Ku80 and DNA-PK; and homology-directed repair (HDR) proteins RPA1, RPA2, and PARP1 (Fig. 2b). MSH2 forms a complex with MSH6 as MutSα, which recognizes mismatched DNA loops including G quadruplexes[43]. During DNA replication when dsDNA separates into ssDNA molecules, the genome may form larger G4 loops, particularly in centromere and telomere regions[44,45]. Therefore, we hypothesize that MutSα may bind to such loop structures and recruit DNA2 to resolve them. MSH2 and MSH6, which in complex is termed MutSα, recognizes DNA mismatches, including those within DNA secondary structures[37]. MutSβ, formed by MSH2 and MSH3, was recently shown to bind to G4-containing loop structures[46]. Using co-IP and western blot analyses, we confirmed that MutSα components MSH2 and MSH6, but not MSH3 co-IPed with DNA2 (Fig. 2c) and that DNA2 co-IPed with MSH2 as well (Fig. 2d), demonstrating that DNA2 interacts with MutSα. Furthermore, using biotin-labeled G4 oligos and streptavidin-coated magnetic beads, we showed that MutSα, but not MutSβ could bind G4 DNA substrates in vitro (Fig. 2e). We additionally tested varying concentrations of MutSα in the presence of G4B beads to determine the binding constant of MSH2 (48 nM) and MSH6 (43 nM) (Fig. 2f–h). In addition, we examined the association of MSH2 and G4 in WT MEFs and found that MSH2 localized to G4 ~ 15% more frequently than in a random distribution of points (Fig. 2i, j). These findings suggest that MSH2 may play a role in regulating the G4 excision function of DNA2. To determine if MutSα or MutSβ stimulates DNA2-mediated cleavage of G4s, we tested G4 cleavage by DNA2 in the absence and presence of MutSα or MutSβ. We found that MutSα, but not MutSβ, stimulated DNA2 to cleave G4 substrates in vitro (Fig. 2k, l).

To determine the importance of MutSα in G4 resolution in cells, we analyzed G4 levels in WT, $MSH2^{-/-}$, and $MSH6^{-/-}$ MEFs. Like DNA2 inhibition or deficiency, $MSH2^{-/-}$ and $MSH6^{-/-}$ cells had significantly

more G4 foci than the WT cells (Fig. 3a, b). To understand how MSH2 regulates DNA2 activity for G4 resolution, we compared the colocalization of DNA2 and G4 in WT and $MSH2^{-/-}$ MEF cells (Fig. 3c). We compared the relative frequencies of co-localization compared to a random distribution of points. We observed that DNA2 was slightly more enriched at G4 spots in WT cells, with a ~20% higher relative co-localization compared to $MSH2^{-/-}$ cells (Fig. 3d). When we analyzed the DNA2–G4 colocalized foci with telomeres, we found that DNA2-G4 foci were found ~7x more frequently at telomeric regions in WT cells than in $MSH2^{-/-}$ cells (Fig. 3e) while the difference in all other genome regions is not as substantial (Fig. 3d). Additionally, we found that MSH2 and MSH6 increased binding of DNA2 to G4 in vitro (Fig. 3f). These findings provide strong evidence that MSH2 facilitates the recruitment of DNA2 at G4-enriched telomeric sites.

The findings that MutSα stimulates DNA2-mediated cleavage of G4s and facilitates binding suggest that MutSα plays an important role in G4 excision to facilitate DNA replication at DTRs such as telomeres. To determine the biological significance of MutSα in telomere replication, we carried out SMARD assays on WT and $MSH2^{-/-}$ MEFs. Like $DNA2^{+/-}$ MEFs, $MSH2^{-/-}$ MEFs had significantly shorter replicated telomeres and showed more replication stalling at telomeres (Fig. 3h, i). The median length of replicated telomeres was 4.4 μm in $MSH2^{-/-}$ MEFs, compared to 8.7 μm in the WT (Fig. 3h). Fifty two percent of telomeres showed fork stalling (20% partial and 32% adjacent) in $MSH2^{-/-}$ MEFs, compared to ~20% in the WT (Fig. 3i). Consistently, with telomere FISH, we found that $MSH2^{-/-}$ MEF cells had significantly more fragile telomeres. $MSH6^{-/-}$ MEF cells also exhibited an increase in telomere fragility, at a rate statistically indistinguishable from $MSH2^{-/-}$, confirming the role of MutSα in DNA2-mediated G4 repair at telomeres (Fig. 3j, k).

## G4 stabilizing compounds block FANCJ-driven G4 unwinding, causing polymerase pausing, but do not affect DNA2-mediated G4 cleavage

Next, we sought to investigate the impact of environmental chemical compounds (ECCs) on G4 resolution in cells, particularly under a DNA2 haplo-insufficient or MSH2 knockout background. To identify possible G4-binding ECCs, we used our in-house-developed, structure-based, virtual ligand screening pipeline to screen a naturally occurring chemical library for potential binding to G4s (Supplementary Fig. 7). Potential G4-binding ECCs were defined as those with the highest docking scores (Supplementary Table 1). From our docking screen, protoporphyrin IX (PPIX), a porphyrin derivative, and perylene derivatives pigment red 123 (PR123) (Supplementary Fig. 7a) and PIPER (Supplementary Fig. 7b) were among the ECCs with the highest potential for binding to G4. Consistently, PIPER and PPIX were both previously reported as potent G4-binding chemicals[29,47,48]. In addition to the known G4-binding compounds, we found bleomycin, polyazo dye, aflatoxin B1, and capreomycin had high docking scores (model docking of ECCs to G4, Supplementary Fig. 7c). Therefore, we tested the impact of the known G4-binding perylene derivative PIPER, the porphyrin derivative TMPyP4, and the aminoglycoside antibiotic, capreomycin, on G4 resolution via helicase-driven G4 unwinding or DNA2-mediated G4 cleavage.

To determine the impact of these G4-stabilizing ECCs on G4 resolution via unwinding, we carried out Polδ-catalyzed primer extension on DNA templates which contained either a non-G4 forming sequence or a G4 forming sequence in the middle of the template (Supplementary Fig. 8a). Polδ effectively extended the primer on the non-G4 forming DNA template, but it paused before the G4 forming sequence upon an increase in the $K^+$ concentration, which is a G4-stabilizing ion, further confirming that this effect is due to the presence of G4 (Supplementary Fig. 8b). Meanwhile, FANCJ, a typical helicase that unwinds G4s, relieved the pausing effect (Supplementary Fig. 8c, d). Conversely, we found that the known G4-inducing/stabilizing compounds PIPER, TMPyP4, NNMP enhanced the pausing effect

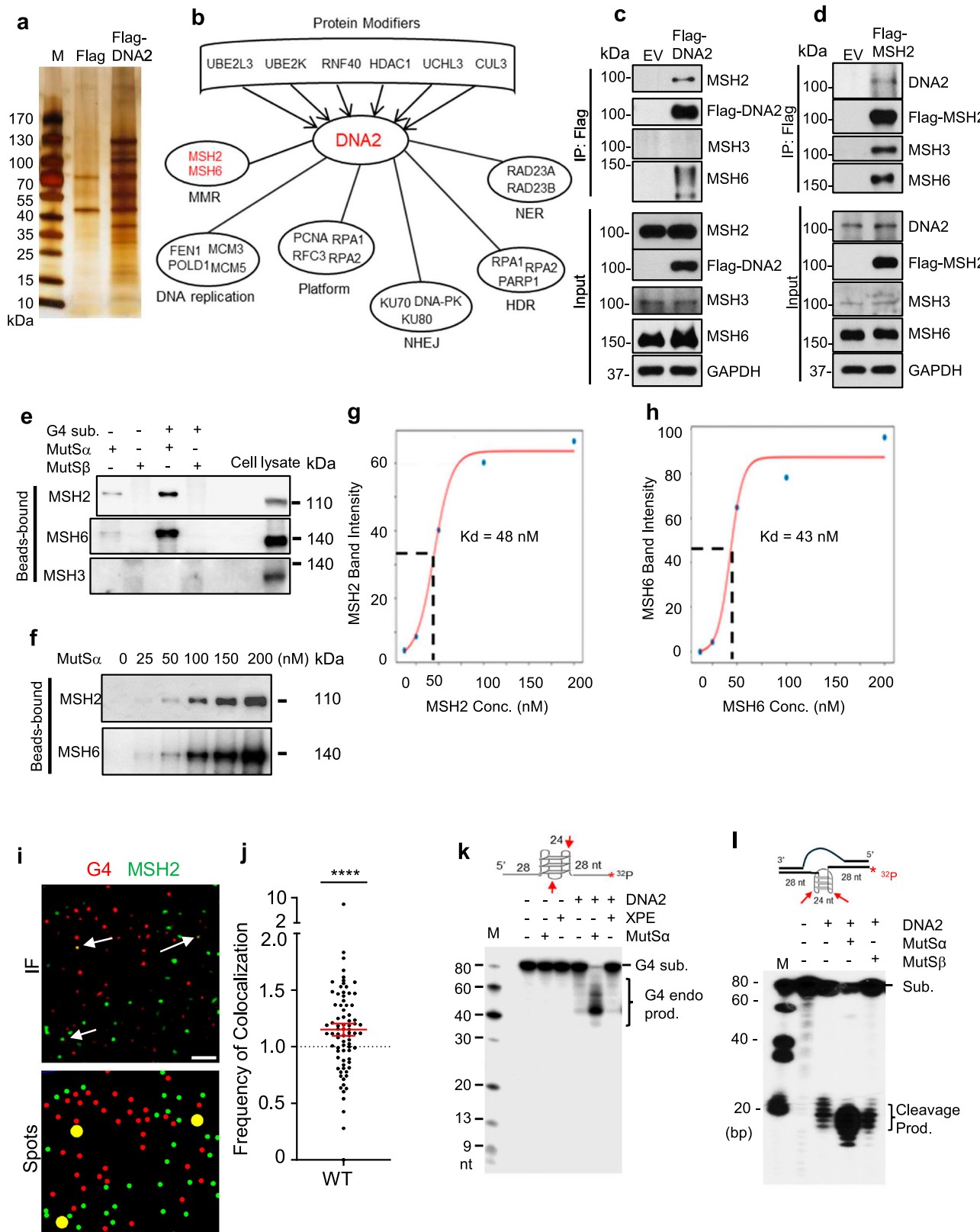

and reduced the fully extended product, but they did not affect Polδ extending on the non-G4 template (Fig. 4a and Supplementary Fig. 9a, b). Consistent with our suggestion that capreomycin is a G4-binding ECC, we observed that it can also enhance Polδ pausing at the G4-forming sequence (Fig. 4b). We further found that PIPER inhibited FANCJ unwinding G4 (Fig. 4c), while capreomycin had a more modest

effect on FANCJ activity (Fig. 4d). We observed that similar to TMPyP4[31], neither PIPER nor capreomycin affected DNA2-mediated cleavage of the G4 DNA substrate (Fig. 4e, f). We further analyzed the impact of G4 stabilizing compounds on the accumulation of G4 foci in WT, *DNA2*[+/−], and *MSH2*[−/−] MEF cells. Treatment of WT MEFs with PIPER increased the accumulation of G4 foci compared to the untreated WT

**Fig. 2 | G4 binding protein complex MutSα stimulates DNA2-mediated G4 cleavage. a** Representative silver staining SDS-PAGE from three replicates showing the pulled-down proteins by Flag-tag M2 beads in whole cell extracts from 293T cells transfected with 3x-Flag human DNA2; **b** DNA replication and repair proteins co-pulled down with 3x-Flag-tagged DNA2; **c, d** Representative co-IP and western blot analysis from three replicates of DNA2 interaction with MSH2 and MSH6; **e** Representative blot from three experiments of MutSα (50 nM) and MutSβ (50 nM) incubated 2 pmol substrates/μl beads. Bead-bound MutSα (MSH2 and MSH6) or MutSβ (MSH2 and MSH3) were detected by western blot; **f** Representative blot from three experiments with varying concentrations of MutSα (0, 25, 50, 100, 150, 200 nM) incubated with 1 μl streptavidin magnetic bead-linked G4 substrates. Bead-bound MSH2 or MSH6 was analyzed by western blot. Binding curves: the x-axis is MSH2 (**g**) or MSH6 (**h**) concentration and the y-axis is relative intensity of MSH2 (**g**) or MSH6 (**h**). Kd was determined by sigmoid curve fitting and corresponds to the concentration at which 50% of MSH2 or MSH6 bind to G4 substrates. Concentrations from each lane in **f** were used to determine the binding affinity of MSH2 (48 nM) and MSH6 (43 nM) to G4B oligo; **i** AIRYSCAN confocal microscopy of G4s; MSH2-G4 co-immunofluorescence staining in MEFs (top panel). The bottom panel shows identified spots of MSH2 and G4. Scale bar = 1 μm; **j** Frequency of co-localization of MSH2 and G4. Red line indicates mean ± SEM. P-value was calculated using a two-sided paired t-test (n = 79 cells, p < 0.0001); **k** The representative denaturing PAGE image shows DNA2 (16 nM) cleaving the $^{32}$P-labeled G4 substrate (top) in the absence or presence of MutSα (50 nM) or XPE (100 nM); **l** The representative denaturing PAGE image shows cleavage of the $^{32}$P-labeled G4 bubble substrate (top) by the DNA2 (16 nM) in the absence or presence of MutSα (50 nM) or MutSβ (50 nM). Source data are provided as a Source data file. Created in BioRender. Zhou, T. (2025) https://BioRender.com/2d65a6r.

MEFs (Fig. 4g, h). DNA2 haploinsufficiency or MSH2 deficiency and 1 μM PIPER treatment potentiated each other in the accumulation of G4 foci. Capreomycin at 15 μM, like PIPER at 1 μM, caused an increase in G4 foci in WT MEFs and had a significant potentiation effect with DNA2 haploinsufficiency or MSH2 deficiency (Fig. 4g, h). We also noted that the use of G4 stabilizing compounds increased the apparent frequency of DNA2 and G4 co-localization, as PIPER-treated cells showed a greater frequency of co-localization of DNA2 and G4 than that found in untreated cells (Supplementary Fig. 10), suggesting that DNA2 is particularly important when helicase-mediated G4 resolution is inhibited.

### G4-stabilizing compounds PIPER and capreomycin delay DNA replication at telomeres

To determine the impact of G4 accumulation caused by G4 stabilizing compounds on telomere replication in WT, *DNA2*$^{+/-}$, and *MSH2*$^{-/-}$ MEFs, we treated the cells with the G4-inducing/stabilizing compounds PIPER or capreomycin and performed SMARD to analyze replication dynamics at the telomeres of these MEFs. We found that both PIPER and capreomycin caused the slowing or stalling of replication at telomeres (Fig. 5a). Exposure to PIPER or capreomycin reduced the median length of replicated telomeres from 8.7 μm (Fig. 1g) to 5.7 μm and 5.4 μm, respectively, in WT cells (Fig. 5b). Similarly, PIPER and capreomycin reduced the median length of replicated telomeres from 4.1 μm (Fig. 1g) to 2.7 μm and 2.5 μm, respectively, in *DNA2*$^{+/-}$ cells (Fig. 5b). However, PIPER and capreomycin did not cause a reduction in the median length of replicated telomeres in *MSH2*$^{-/-}$ MEFs (Figs. 3h and 5b). Meanwhile, we observed that PIPER and capreomycin increased the frequency of stalled replication forks at or in telomeres from 22% to 43% (Fig. 5c) in WT cells and increased the frequency from 50% to 55% and 59%, respectively, in *DNA2*$^{+/-}$ cells (Fig. 5c). However, PIPER and capreomycin did not cause further fork stalling at the telomere in *MSH2*$^{-/-}$ MEF cells (Figs. 3i and 5c).

Consistent with the result of telomere synthesis impairment due to exposure to PIPER and capreomycin, using telomere FISH we observed that PIPER or capreomycin exposure alone was sufficient to cause an increased frequency of telomere abnormalities, including fragility, loss, and/or fusion in WT cells (Fig. 6a–d). *DNA2*$^{+/-}$ MEF cells displayed significantly more telomere abnormalities than the WT MEF cells under normal culture conditions, and exposure to PIPER or capreomycin caused additional telomere defects (Fig. 6a, b). Similar to *DNA2*$^{+/-}$ MEF cells, *MSH2*$^{-/-}$ MEF cells exhibited more sensitive to PIPER and capreomycin with a corresponding increase in telomere fragility, loss or fusion (Fig. 6c, d). It is worth noting that *DNA2*$^{+/-}$ cells were more sensitive to PIPER or capreomycin than *MSH2*$^{-/-}$ cells (Fig. 6a–d). It is possible that in the absence of MSH2, other proteins may facilitate DNA2 mediated G4 excision, albeit at much less efficiency.

## Discussion

Our current studies define the biological importance of DNA2 nuclease-mediated G4 excision. We show that *DNA2* gene deficiency or treatment with a DNA2 inhibitor results in G4 accumulation in MEFs and using super resolution microscopy demonstrate that DNA2 co-localizes with G4 in cells at sub-diffraction distances. We further determine the impact of G4-inducing/stabilizing chemicals on G4 excision as opposed to G4 unwinding, the well-studied G4 resolution mechanism. We show that FANCJ, a representative helicase that unwinds G4[25], effectively assists Polδ in passing through the G4-forming sequence. However, G4-stabilizing ECCs, including PIPER and TMPyP4, inhibit FANCJ's G4 unwinding activity, leading to Polδ pausing at G4s. Meanwhile, G4 stabilization by PIPER or other G4-stabilizing ECCs has little effect on DNA2-mediated G4 excision. It suggests that two different G4 resolution mechanisms may function under different conditions. Under normal physiology, helicase driven G4 unwinding is the primary pathway, and G4 excision is an important pathway for efficient G4 resolution. However, DNA2-mediated G4 excision becomes the primary pathway for G4 resolution when cells are exposed to G4-stabilizing ECCs as helicases are no longer able to resolve these structures.

Given the nature of G-rich sequences to spontaneously form stable G4s or G4-like structures, G-rich sequences typified by telomere TTAGGG repeats are considered DTR regions. We previously used *DNA2*$^{+/-}$ MEFs to show that DNA2 deficiency caused fragile telomeres[31], indirectly implicating DNA2 in telomere replication. In our current studies, using the single-molecule-based SMARD assay, we successfully assessed DNA replication at telomeres. We demonstrate that DNA2 deficiency or inhibition results in stalled replication forks at telomeres and telomere instabilities, especially in the presence of G4-stabilizing ECCs, including PIPER and a previously uncharacterized chemical stabilizer, capreomycin, in mammalian cells.

We further identify MutSα as a G4-binding protein complex to facilitate G4 resolution via DNA2-mediated G4 excision. We observed that MutSα binds G4 in vitro and in cells. Furthermore, using spatial analysis algorithms, we show that co-localization of DNA2 with G4 is dependent on MSH2, both throughout the genome and especially at telomeres. In vitro, MutSα interacts with DNA2 and stimulates DNA2 to cleave G4 structures, though MutSβ had no effect on DNA2 nucleolytic activity. Consistent with the biochemical data, MSH2 and MSH6 knockout, similar to DNA2 deficiency, causes G4 foci accumulation. Since G4 impedes replication fork progression[49], it is not surprising to see that proteins involved in G4 resolution are necessary to maintain replication dynamics through regions enriched in G4 structures (i.e., telomeres). As a result, cells likely require the use of alternative pathways to maintain telomeres when these proteins are missing, such as ALT telomere lengthening[50], which is in agreement with previous research that has shown that cells deficient in MutSα are prone to undergo ALT telomere maintenance[51], and that DNA2 restricts ALT[52].

Due to the problems in telomere replication, we show that cells lacking DNA2 are more prone to genome rearrangements, such as telomere fusion or sister chromatin exchange[31]. We additionally

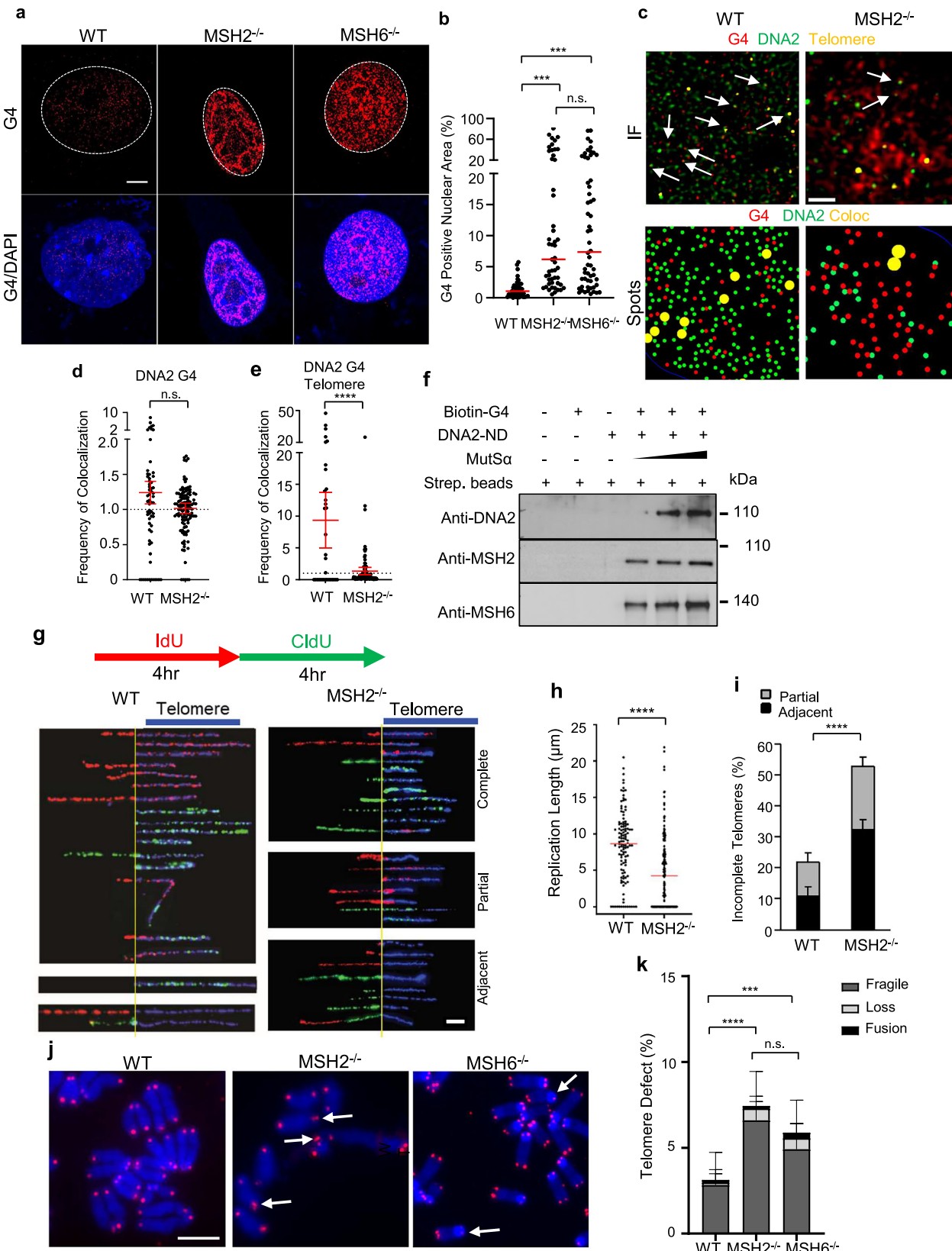

describe the role of environmentally contaminating compounds on telomere maintenance and the requirement for functioning DNA2 and MSH2 to resolve the impact of these compounds. While these compounds alone are sufficient to slow down normal replication of telomeres, the combination of DNA2 and/or MSH2 deficiency leads to a greater occurrence of fragile telomeres and other telomere defects. In our analysis of the recruitment of DNA2 to G4 in telomere regions via IF and pulldown/western blotting, we found that absence of MSH2 significantly reduces DNA2 recruitment to G4 structures. MutSα facilitates the recruitment of DNA2 onto the G4 structures. Similarly, in DNA2+/− cells, there is insufficient DNA2 protein to be recruited onto the G4 structures. Therefore, the absence of MSH2 or

**Fig. 3 | MSH2 deficiency results in G4 accumulation, defects in telomere replication. a, b** G4 immunofluorescence staining in WT, MSH2$^{-/-}$, and MSH6$^{-/-}$ MEFs. **a** shows the representative AIRYSCAN confocal microscopy images of G4s, and **b** shows the quantification of G4 in WT, MSH2$^{-/-}$, and MSH6$^{-/-}$ MEFs (n = 45, 51, 58 cells). The red line indicates median. All p-values were calculated using one-way ANOVA (WT vs. MSH2$^{-/-}$ p = 0.0002, WT vs. MSH6$^{-/-}$ p = 0.0001). Scale bar = 5 μm; **c** AIRYSCAN jDCV rendering of co-immunofluorescence of DNA2 and G4, and telomere FISH in WT and MSH2$^{-/-}$ MEFs (top panels); spots of DNA2 and G4 and spots of DNA2 within 100 nm of a G4 spot (bottom panels); **d** Calculated relative co-localization frequency of DNA2 and G4 in WT and MSH2$^{-/-}$ cells (n = 64, 109 cells). Red line indicates mean ± SEM. P-value was calculated using the two-sided Student's *t*-test; **e** Relative co-localization frequency of DNA2 and G4 with telomeres in WT and MSH2$^{-/-}$ cells (n = 33, 104 cells). Red line indicates mean ± SEM. P-value was calculated using the two-sided Student's *t*-test (p = 0.0008); **f** Representative blot from three independent experiments showing increasing DNA2 binding to G4 oligos with increasing MutSα; **g–i** SMARD analysis of WT and MSH2$^{-/-}$ MEF cells. Scale = 5 μm. **g** Representative microscopy images of the replicated telomeres in WT or MSH2$^{-/-}$; **h** Quantification of the length of replicated telomeres in different cells (n = 118, 156 fibers). The red line indicates median. P-value was calculated using a two-tailed Student's *t*-test (p < 0.0001); **i** Percentage of partial telomere replication and replication fork stalling adjacent to telomeres in WT or MSH2$^{-/-}$ cells (n = 119, 167 fibers). Bar indicates mean and SD. P-value was calculated using a two-tailed Chi-squared test (p < 0.0001); **j** Telomere FISH images showing telomere abnormalities in the WT, MSH2$^{-/-}$, and MSH6$^{-/-}$ MEFs; **k** Quantification of telomere defects in WT, MSH2$^{-/-}$, and MSH6$^{-/-}$ MEFs (n = 29, 15, 34 spreads). Bars indicate mean and standard deviation. All p values were calculated using the two-way ANOVA (WT vs. MSH2$^{-/-}$ p < 0.0001; WT vs. MSH6$^{-/-}$ p = 0.0002). *p < 0.05, **p < 0.01, ***p < 0.001, and ****p < 0.0001. Source data are provided as a Source data file.

haploinsufficiency of DNA2 may cause similar cellular phenotypes of shortened telomere length and abnormal telomere frequency.

In summary, our current studies provide several lines of evidence that DNA2-mediated G4 excision repair is important for the resolution of G4 in mammalian cells and efficient and faithful telomeric DNA replication, especially in the presence of G4-stabilizing compounds, which suppress G4 unwinding by helicases. Furthermore, we define the DNA repair protein complex MutSα that recognizes G4s, recruits DNA2, and stimulates its cleavage of G4s. G4 accumulation causes replication stress, which is a major source of genome instability. We demonstrate that DNA2 or MSH2 deficiency in MEF cells causes telomere instability. Therefore, DNA2 haploinsufficiency that causes defective G4 excision leads to genome instability and contributes to tumorigenesis[31]. On the other hand, the results from our current studies also have implication in cancer cell survival in response to anti-cancer treatment. G4 induction and stabilization by drugs such as CX5641 have been used as a strategy to create overwhelming replication stress to kill cancer cells[53]. Robust G4 excision is a hurdle for the effective killing of cancer cells by G4-stabilizing drugs. Simultaneous induction of G4 formation and inhibition of G4 excision could be an effective combination for future cancer therapy.

## Methods
### DNA2 pulldown and mass spectrometry
3x-Flag-tagged DNA2 was expressed or pulled down with anti-Flag tag M2 beads as previously described[30,31,54]. Briefly, cells transfected with the empty vector (control) or the vector encoding 3x-Flag-DNA2 were lysed by brief sonication in an immunoprecipitation (IP) buffer containing 50 mM HEPES-KOH (pH 7.4), 150 mM NaCl, 0.1% NP40, 10% glycerol, and 1× protein inhibitor cocktail (ThermoFisher). After centrifugation (20,000 × *g*, 15 min, 4 °C), the clear supernatant was incubated with anti-Flag M2 magnetic beads (Sigma, M8823) overnight. The beads were washed with a wash buffer containing 50 mM HEPES-KOH (pH 7.4), 500 mM NaCl, 0.1% NP40, and 10% glycerol, and 1x protein inhibitor cocktail once and the IP buffer once. The DNA2 protein complexes were eluted with 3x-Flag peptide (250 μg/ml). The eluted protein complexes and the control were run into a 4–15% Mini-PROTEAN® TGX™ Precast Protein Gel (Bio-Rad) and stained with a silver staining kit or Coomassie brilliant blue (ThermoFisher Scientific). All protein bands were de-stained and excised, followed by in-gel digestion, extraction, and LC/MS analysis using a Thermo Scientific Orbitrap Fusion Mass Spectrometer following the supplier's instruction. The proteins were identified using both Proteome Discoverer Software with Sequest (Version 2.0) and the Mascot algorithm (Mascot 2.5.1). All identified proteins have been listed in the ProteomeXchange repository with the identifier PXD059843 and analyzed by enrichment relative to the empty vector control.

### Immunoprecipitation and western blot
Immunoprecipitation (IP) was carried out as previously described[54]. Briefly, cells were lysed in a buffer containing 0.5% Nonidet P-40, 50 mM Tris-HCl, 0.1 mM EDTA, 150 mM NaCl, and proteinase inhibitors. Whole-cell lysates were incubated with the antibody against a specific protein and Pierce™ Protein A/G Magnetic Beads (Thermo-Fisher Scientific). The beads were washed with a wash buffer containing 0.5% Nonidet P-40, 50 mM Tris-HCl, 0.1 mM EDTA, 500 mM NaCl, and proteinase inhibitors. The IPed proteins were eluted with 2x SDS loading buffer and analyzed by western blot following the standard western blot protocol. The following antibodies were used: DYKDDDDK (Cell Signaling Technology, 8146, 1:1000), DNA2 (Millipore Sigma, PA5-115131, 1:1000), MSH6 (BD, 610918, 1:1000), MSH3 (Santa Cruz, sc-5686, 1:1000), MSH2 (Cell Signaling Technology, D24B5, 1:1000), GAPDH (Cell Signaling Technology, 14C10, 1:1000), Donkey anti-goat (Santa Cruz, sc-2020, 1:2000), and Goat anti-rabbit (Cell Signaling Technology, 7074, 1:2000).

### G4 binding assay
The G4 DNA substrate binding by MutSα or MutSβ or DNA2 was measured using a protocol that was modified from a published method[55]. The biotinylated G4 DNA substrate was immobilized onto the Streptavidin magnetic beads (Pierce) by incubation in buffer A (10 mM Tris-HCl, pH 7.5, 1 mM EDTA, and 100 mM KCl) at 25 °C for 2 h. The beads were washed twice with the same buffer solution and blocked with 3% BSA in buffer A at 25 °C for 30 min. After washing the beads with buffer A once, the beads were suspended in buffer B containing 30 mM HEPES-KOH, pH 7.5, 8 mM magnesium acetate, 0.2 mg/ml BSA, and 1 mM DTT. Specific amounts of indicated protein were incubated with the empty beads (control) or G4 DNA coated beads at 25 °C for 30 min. The beads were then washed three times with buffer C (10 mM Tris-HCl, pH 7.5, 1 mM EDTA, and 150 mM KCl. 0.01% Tween 20). The bead bound proteins were eluted with 2x SDS loading buffer containing 200 mM DTT and boiled in 2× protein sample buffer (100 mM Tris-HCl, pH 6.8, 200 mM DTT, 4% SDS, 0.2% bromophenol blue, and 20% glycerol). The protein supernatant was separated by 10% sodium dodecyl sulfate−polyacrylamide gel electrophoresis (SDS−PAGE) and analyzed by western blot.

### DNA2 nuclease assay
To make the G4 substrate, the G4B oligo was labeled with FAM at the 3′ end or labeled with $^{32}$P using DNA terminal transferase (Sigma) following the supplier's instruction. The oligo sequences are specified in Supplementary Table 2. The G4 structure substrate was made according to a previously described protocol[31]. The formation of G4, which migrates faster than the corresponding non-G4, was confirmed by non-denaturing PAGE (Supplementary Fig. S2). To assay the nuclease activities, 16 nM (20 ng) DNA2 protein was incubated with 50 nM DNA substrate in a reaction buffer containing 20 mM Tris-HCl

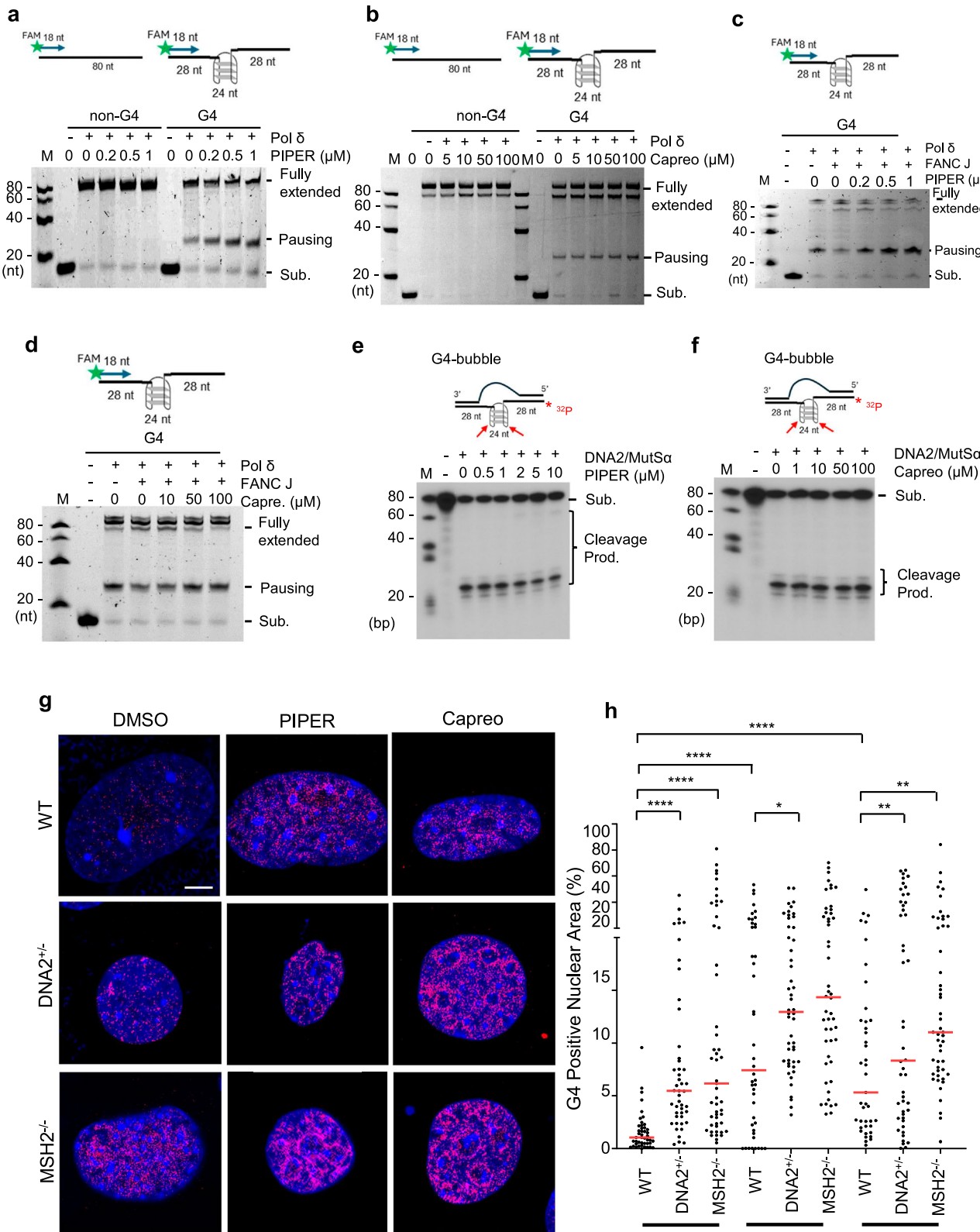

(pH 7.5), 8 mM MgCl$_2$, 1 mM DTT, 100 mM KCl, and 0.1 mg/ml BSA at 37 °C for 30 min. The reactions were stopped with the addition of an equal volume of a 2x STOP solution (10 mM EDTA in 95% formamide) at 95 °C for 15 min. The DNA substrates and products were analyzed with a 15% denaturing PAGE gel and visualized by a Typhoon FLA9500 scanner or by radioautography.

## Reconstituted G4 excision and gap filling assay

To reconstitute G4 excision by DNA2 and subsequent gap filling by Polδ or Polβ, 16 nM DNA2 was incubated with 10 nM G4 bubble structure, which is prepared by annealing the G4B oligo and the G4B-T oligo at 1:1 ratio, in a reaction buffer containing 20 mM Tris-HCl (pH 7.5), 8 mM MgCl$_2$, 1 mM DTT, 20 mM KCl, and 0.1 mg/ml BSA at 37 °C

**Fig. 4 | G4-stabilizing ECCs inhibit FANCJ unwinding G4, but not DNA2-mediated cleavage of G4 in vitro. a, b** Representative primer extension from three replicates by Polδ (20 nM) on the non-G4 or G4 template in the presence of increasing concentrations of G4-stabilizing ECCs PIPER (0.2, 0.5, 1 μM) (**a**) and capreomycin (5, 10, 50, 100 μM) (**a**). **b** The top shows the diagram of the Polδ (20 nM)-catalyzed primer extension on a DNA template without or with a G4-forming sequence. The primer was labeled with FAM on the 5′ end. The bottom shows the representative denaturing-PAGE image of the primer extension assay; **c, d** Representative primer extension from three replicates on the G4 template in the absence or presence of 5 nM FANCJ with or without 0.2, 0.5, and 1 μM PIPER (**c**) or 10, 50, and 100 μM capreomycin (**d**); **e, f** Representative blot from three replicates showing the cleavage of a FAM³²P-labeled G4 substrate (50 nM) with DNA2

(16 nM) and MutSα (50 nM) in the absence or presence of increasing concentrations of PIPER (0.5, 1, 2, 5, 10 μM) (**e**) or capreomycin (1, 10, 50, 100 μM) (**f**); **g, h** G4 immunofluorescence staining in WT, DNA2⁺/⁻, and MSH2⁻/⁻ MEFs. **g** The representative AIRYSCAN confocal microscopy images of G4s, and (**h**) shows the quantification of G4 in WT, DNA2⁺/⁻, or MSH2⁻/⁻ MEFs without or with treatment with G4 stabilizers PIPER or capreomycin (n = 45, 47, 51, 44, 50, 50, 41, 50, 48 cells). Results are from at least 3 biological replicates. Red lines indicate median. All p-values were calculated using a two-tailed Student's t-test (WT NT vs. WT Capreo p < 0.0001; WT NT vs. WT PIPER p < 0.0001; WT Capreo vs. DNA2+/− Capreo p = 0.021; WT PIPER vs. DNA2+/− PIPER p = 0.005; WT PIPER vs. MSH2⁻/⁻ PIPER p = 0.005). Scale bar = 5 μm. Source data are provided as a Source data file. Created in BioRender. Zhou, T. (2025) https://BioRender.com/d58fiu1.

for 30 min. Polδ (20 nM) or Polβ (250 nM) and 5 μci [α–³²P] dTTP and 50 μM each of dCTP, dGTP, and dATP were added to the reactions. The reactions were incubated for 2.5, 5, 10, 20 min. The reactions were stopped with addition of an equal volume of a 2x STOP solution (10 mM EDTA in 95% formamide) at 95 °C for 15 min. The DNA substrates and products were analyzed with a 15% denaturing PAGE gel and visualized by radioautography.

### G4 staining in cell culture, immunostaining, FISH, and foci quantification

Cells were fixed using 4% PFA in PBS for 15 min at RT, followed by 0.5% Triton X-100 (15 min, RT), and blocking in 5% BSA + 0.1% Tween-20 and Image iT-FX Signal Enhancer. The anti-G4 primary antibody (Sigma, MABE1126, 1:500) was diluted 1:500 in 4% BSA and 0.1% Tween-20 overnight at 4 °C, washed with 0.1% Tween-20, and stained with 1:1000 GAM Alexa Fluor 568 (Thermo, A11031) for 1 h at RT. Cells were counterstained with DAPI and mounted in ProLong Gold (Thermo).

Imaging was performed using the Zeiss LSM 900 with Airyscan 2. Briefly, we zoomed into individual nuclei and scanned with a resolution that a theoretical point spread function would be sampled at least 3×. After acquisition, images were processed using Zeiss's Airyscan Joint Deconvolution algorithm at 10 iterations to ensure maximum resolution.

To quantify foci, we used the particle finder algorithm on ImageJ (1.54 f). We used a consistent threshold across all imaging conditions and measured the area of foci divided by the area of the nucleus over at least 40 cells and at least 3 biological replicates per condition. For Co-IF experiments, Rabbit anti-DNA2 (Invitrogen, PA5-115131, 1:1000) and Rabbit anti-MSH2 (Cell Signaling Technology, D24B5, 1:1000) were labeled in cells using primary antibodies. Goat anti-rabbit 647 (Invitrogen, A-21244, 1:1000) was used. For imaging experiments where FISH was used, cells were dehydrated using an ethanol series dehydration and labeled with a TelG-Alexa 488 telomere probe (PNA Bio) at 37 °C for 2 h. After rinsing, cells were immunostained for G4 and the appropriate protein.

### Co-localization analysis

To quantify the extent of co-localization and generate images, we used Imaris (Oxford Instruments, version 10.2) software to analyze IF images. We used a co-localization algorithm similar to previous studies[56]. Briefly, first, we determined nuclear area by drawing a surface based on DAPI intensity. Next, we identified spots of various features (DNA2, MSH2, G4, or telomere) using the spot finding algorithm in the software with a consistent quality factor across images. Only spots within the nucleus were considered. Then we used nearest-neighbor analysis to find the number of spots that were within 100 nm (roughly the resolution limit of the joint deconvolution algorithm) of each other to classify as co-localized. Because we only used measured distances below 100 nm, we safely neglected edge corrections. We then normalized the number of co-localized spots to the expected frequency of

co-localization, which is based on the density of spots and our search radius. This gave us relative co-localization frequencies where 1 is the expected co-localization from a random distribution of points. Values above 1 were considered significantly co-localized.

### Single molecule analysis of replicated DNA

SMARD was carried out according to the protocol developed by Norio and Schildkraut[36]. Cells were grown in DMEM with 10% FBS (Gibco) and pen/strep (Gen Clone). Subconfluent, dividing cells were successively pulse-labeled with IdU (30 μM, 4 h, Sigma) and then CldU (30 μM, 4 h, Sigma). After labeling, cells were trypsinized, harvested, and cast into 0.5% agarose plugs (TopVision). Cells were lysed at 50 °C in 1% n-lauroylsarcosine, 0.5 M EDTA, and 0.2 mg/ml Proteinase K (Bioland). DNA was digested outside of telomere regions using *SwaI* (NEB) overnight at RT. Digested DNA was then separated by electrophoresis with 0.7% SeaPlaque GTG Agarose (Lonza) in 0.5X TBE buffer. DNA fragments from 15 to 50 kb were excised and the agarose digested with β-agarase (NEB). This DNA was then stretched on 3-aminopropyltriethoxysilane-treated coverslips (Sigma), denatured with NaOH in 70% ethanol and fixed with glutaraldehyde. DNA was hybridized with the TelC biotinylated telomere probe (PNA Bio) at 37 °C for 2 h. After hybridization, DNA was blocked with 5% BSA and stained with Avidin Alexa Fluor 350 (Thermo, A11236, 1:500), Mouse Anti-BrdU (BD, 347580, 1:100) specific for IdU, and Rat Anti-BrdU (Abcam, ab6326, 1:500) specific for CldU. DNA was then stained again with a Goat Anti-Mouse Alexa 568 (Invitrogen, A11031, 1:100), Goat Anti-Rat DyLight 488 (Invitrogen, SA5-10018, 1:50), and biotinylated Goat Anti-Avidin D (Vector Labs, BA-0300-.5, 1:500). Alternating rounds of avidin and anti-avidin were used to amplify the signal of the FISH probe. Coverslips were mounted on slides using ProLong Gold (Thermo) and imaged using the Zeiss Observer II with a 63x objective. Images were processed using ImageJ (1.54 f) and aligned using Adobe Illustrator (29.6.1).

### Telomere fluorescence in situ hybridization

FISH on telomeres was performed as described previously[31]. Cells were treated with colcemid for 5 h, collected by trypsinization, swollen in 0.075 M KCl at 37 °C for 15 min, and fixed in a freshly prepared 3:1 mix of methanol:glacial acetic acid. Slides containing metaphase spreads were then denatured and hybridized to peptide nucleic acid (PNA) probe Cy3-OO-(CCCTAA)3 to visualize telomeres. Slides were then washed and dehydrated in ethanol series. DNA was counterstained with DAPI, and images were taken under Zeiss Observer epi-fluorescence microscope with a 100× objective.

### Polδ extension assay

FAM-labeled primer was annealed to the G4B oligo and the G4 structure was formed in the G4 folding buffer containing 10 mM Tris-HCl (pH 7.5), 50 mM KCl at 95 °C for 5 min, 72 °C for 10 min, and then cooled down to room temperature. To perform the polymerase extension assay, 20 nM (50 ng) Polδ was incubated with 50 nM DNA

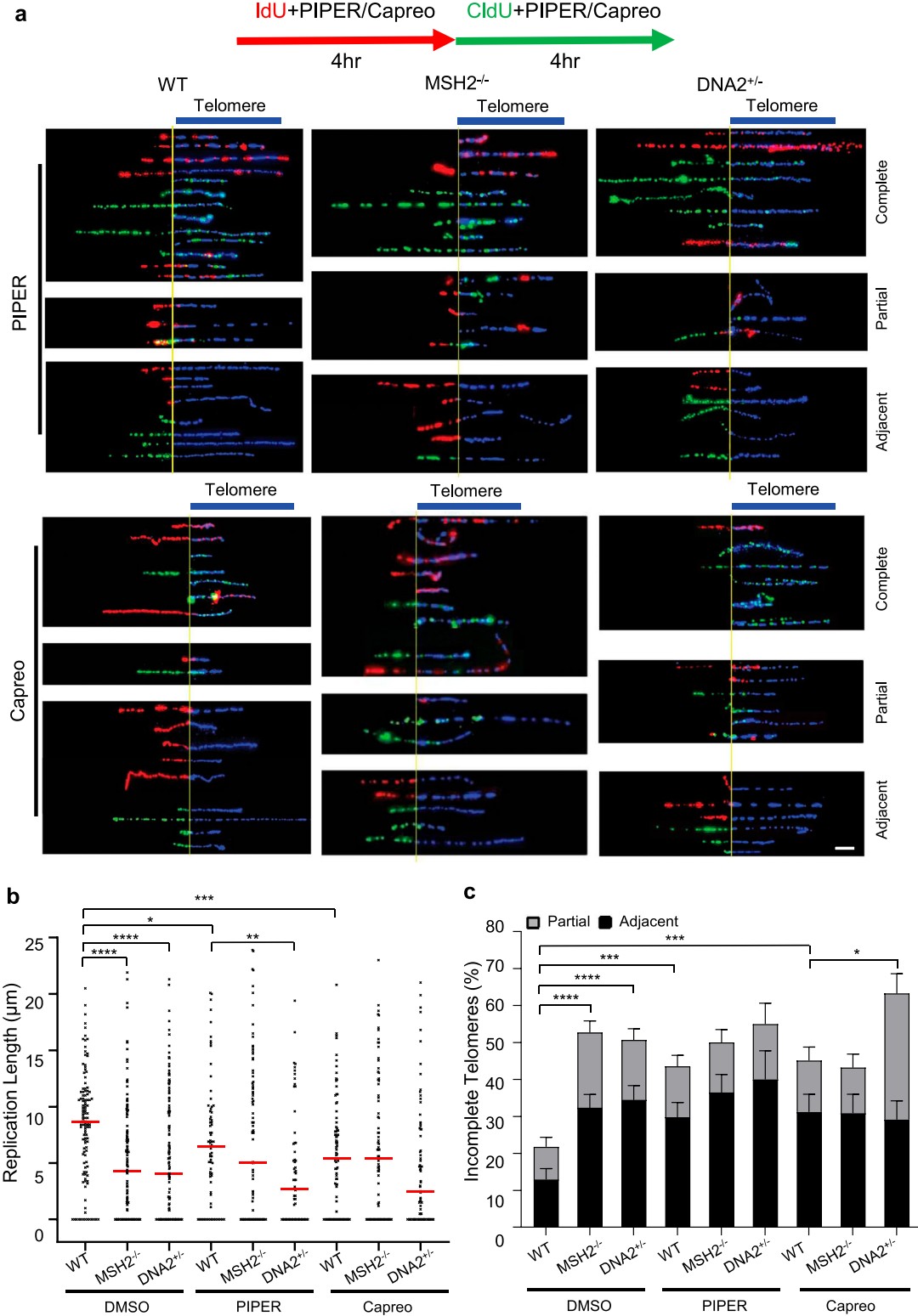

**Fig. 5 | G4-stabilizing ECCs impair DNA replication at telomeres in WT DNA2$^{+/-}$ and MSH2$^{-/-}$ MEF cells. a** SMARD microscopy images of replicated telomeres in WT, DNA2$^{+/-}$, and MSH2$^{-/-}$ MEF cells with or without exposure to PIPER or capreomycin. Scale = 5 µm; **b** Relative length of replicated telomeres (n = 118, 123, 149, 52, 63, 61, 87, 82, 87 fibers). Red lines indicate median. P values were calculated using a two-tailed Mann–Whitney test (WT vs. MSH2$^{-/-}$ p < 0.0001; WT vs. DNA2+/− p < 0.0001; WT DMSO vs. WT PIPER p = 0.0109; WT DMSO vs. WT Capreo p = 0.0001; WT PIPER vs. DNA2$^{+/-}$ PIPER p = 0.046); **c** Percentage of incompletely replicated telomeres (n = 119, 167, 148, 131, 96, 108, 93, 81, 119 fibers). Bars indicate the mean and error bars standard deviation. P values were calculated using a two-tailed Chi-squared test (WT vs. WT Piper p = 0.0004; WT vs. WT Capreo p = 0.0005; WT Capreo vs. DNA2$^{+/-}$ Capreo p = 0.017). *p < 0.05, **p < 0.01, ***p < 0.001, and ****p < 0.0001. Source data are provided as a Source data file.

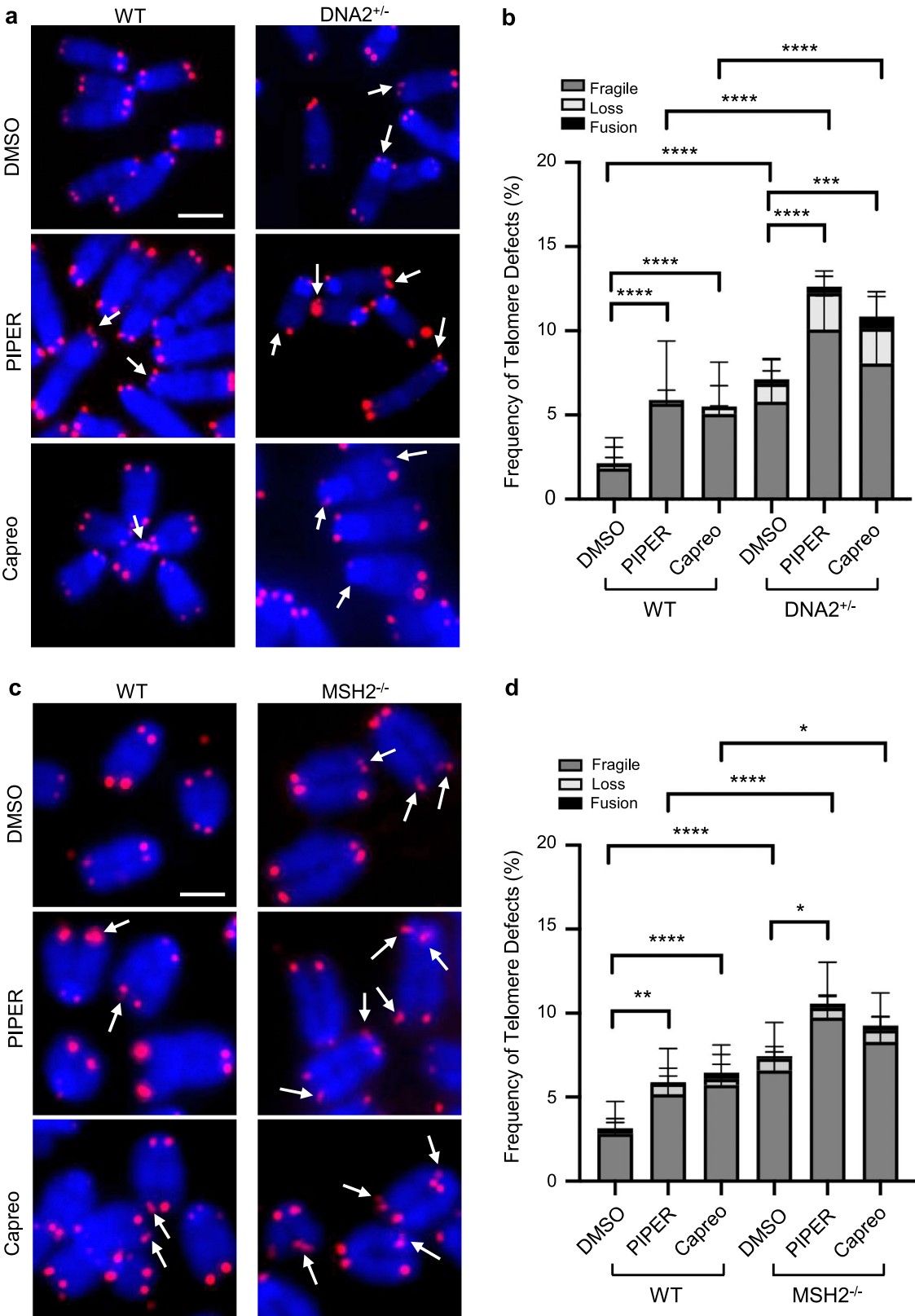

substrates in an extension buffer (20 mM Tris-HCl (pH 7.5), 8 mM MgCl$_2$, 1 mM DTT, 0.10 mg/ml BSA, 20 mM KCl, 250 µM dNTP) with or without the indicated amounts of FANCJ at 37 °C for 30 min. The primer extension reaction was stopped with an equal volume of a 2× STOP solution (10 mM EDTA in 95% formamide) at 95 °C for 15 min. Subsequently, the extensions were resolved on a 15% denaturing PAGE. Gels were imaged with a Typhoon FLA9500 scanner (GE

Healthcare) and quantified with ImageJ (1.54 f). For the polymerase extension assay with G4 stabilizers, varying concentrations of TMPyP4, PIPER, NMMP, and capreomycin were pre-incubated with 50 nM DNA substrate in the extension buffer at room temperature for 5 min, and Polδ was then added to the reactions to a final concentration of 20 nM and the primer extension was carried out and analyzed as described above.

**Fig. 6 | G4-stabilizing ECCs enhance telomere abnormalities in WT DNA2$^{+/-}$ and MSH2$^{-/-}$ MEF cells. a** Telomere FISH images showing telomere abnormalities in the WT and DNA2$^{+/-}$ MEFs treated with PIPER and capreomycin. Scale bar = 5 μm; **b** Quantification of fragile telomeres and shortened telomeres in different cells (n = 57, 44, 56, 24, 21, 22 spreads). All p values were calculated using the two-way ANOVA (WT DMSO vs. WT PIPER p < 0.0001; WT DMSO vs. WT Capreo p < 0.0001; WT DMSO vs. DNA2$^{+/-}$ DMSO p < 0.0001; WT PIPER vs. DNA2$^{+/-}$ PIPER p < 0.0001; WT Capreo vs. DNA2$^{+/-}$ Capreo p < 0.0001; DNA2$^{+/-}$ DMSO vs. DNA2$^{+/-}$ PIPER p < 0.0001; DNA2$^{+/-}$ DMSO vs. DNA2$^{+/-}$ Capreo p = 0.0005); **c** Telomere FISH images showing telomere abnormalities in the WT and MSH2$^{-/-}$ MEFs treated with PIPER and capreomycin. Scale bar = 5 μm; **d** Quantification of fragile telomeres and shortened telomeres in different cells (n = 29, 29, 32, 15, 20, 17 spreads). Bars indicate mean and error bars standard deviation. All p values were calculated using the two-way ANOVA (WT DMSO vs. WT PIPER p = 0.003; WT DMSO vs. WT Capreo p < 0.0001; WT DMSO vs. MSH2$^{-/-}$ DMSO p < 0.0001; WT PIPER vs. MSH2$^{-/-}$ PIPER p < 0.0001; WT Capreo vs. MSH2$^{-/-}$ Capreo p = 0.012; MSH2$^{-/-}$ DMSO vs. MSH2$^{-/-}$ PIPER p = 0.0152);. *p < 0.05, **p < 0.01, ***p < 0.001, and ****p < 0.0001. Source data are provided as a Source data file.

### High-throughput virtual screening to identify G4 stabilizing compounds

We employed a high-throughput virtual screening (HTVS) of 10,449 environmental contaminants to identify potential high-affinity G4 stabilizers. Compound structures were curated from the US Environmental Protection Agency (EPA) CRD2016 database (8708 chemicals from USA industrial and commercial usage), EPA hazardous waste list (724 chemicals), Spin database (1229 industrial chemicals from Nordic countries), Keml database (536 restricted chemicals in Sweden), EPA CCL4 chemicals (96 water-contaminated chemicals), FDA-approved drugs or compounds in clinical studies (6605 molecules from DrugBank), and food toxins and metabolites (210 compounds). Additionally, 36 previously reported G4 stabilizers from the literature and Protein Data Bank (PDB), 37 close structural analogs (similarity cutoff: 0.7), 22 aflatoxin family compounds, and 188 perylene family compounds were included in the analysis. The curated library was docked against G4 crystal structure PDB: 3UYH. Prior to the molecular docking analysis, the curated library and G-quadruplex PDB structure were preprocessed using the LigPrep and Protein Preparation wizards, respectively, within the Maestro docking software (Schrödinger) to resolve common structural issues and adjust the pH to physiological conditions (pH 7.4 ± 1.0).

Potential G4 stabilizers were identified by comparing the curated compounds' docking scores, chemical structures, and binding conformations to the 36 previously reported G4 stabilizers. Docking scores, shown as ΔG (free energy change), have been provided for the top nine potential G4-binding ECCs identified from HTVS (Supplementary Table 1). The most tractable G4 stabilizer candidates were subsequently validated in vivo, leading to the selection of PIPER and capreomycin as potent representative G4 stabilizing compounds for the in vitro experiments within this study.

### Cell lines

WT and DNA2$^{+/-}$ cell lines were generated in our laboratory and previously validated. MSH2$^{-/-}$ and MSH6$^{-/-}$ cell lines were provided by Dr. Winfried Edelmann. Cells were grown in DMEM (Corning) with 10% FBS (Gibco) with 5% $CO_2$.

### Statistical analysis

All statistical analyses were performed using GraphPad Prism (10.4.0) and Microsoft Excel (version 2507).

### Reporting summary

Further information on research design is available in the Nature Portfolio Reporting Summary linked to this article.

## Data availability

Requests for materials should be addressed to the corresponding authors. The mass spectrometry proteomics data have been deposited to the ProteomeXchange Consortium via the PRIDE partner repository with the dataset identifier PXD059843. All other data generated in this study are provided in the main manuscript file and its Supplementary information file. Source data are provided with this paper.

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

## Acknowledgements

We thank Dr. Brian Armstrong at the COH Light Microscopy core facility for his technical support of AIRYSCAN confocal microscopy and SMARD analysis. We thank Eric Zheng for his assistance with data analysis. This work was supported by NIH grants R50 CA211397 to L.Z. and R01 CA085344 to B.H.S. Research reported in this publication includes work performed using City of Hope shared resources supported by the National Cancer Institute of the National Institutes of Health under award number P30 CA033572. This research was also supported in part by the Intramural Research Program of the NIH, National Institute on Aging.

## Author contributions

A.F., T.Z., Y.L., N.L., C.S., H.L., H.Y., L.S. M.Z., and G.S. conducted biochemical and cellular experiments. S.E. conducted in vivo experiments. J.H. did the virtual ECC screening for G4 stabilizers. J.A.S. and R.M.B. provided purified recombinant FANCJ protein and contributed to the design of helicase-relevant experiments. V.G. and W.C. helped with telomere FISH experiments and provided technical advice on data analysis. S.K., D.Z., and C.S. provided SMARD protocols and technical advice. W.E. and N.S. provided MSH2−/− and MSH6−/− MEF cells. G.M.L. provided the purified MutSα and MutSβ protein complex. M.Y.W.T. L. and S. Z. provided the monoclonal antibody used to purify the recombinant DNA polymerase delta protein complex. L.Z. designed biochemical and cellular experiments, analyzed the data, and wrote the manuscript. B.S. supervised the entire project, designed, and coordinated most experiments, provided inputs, and finalized the manuscript.

## Competing interests

The authors declare no competing interests.
