## [Transparent Peer Review file · Nature Communications]

DNA2 and MSH2 cooperatively repair environmental compound stabilized G4 for efficient telomere replication

Corresponding Author: Professor Binghui Shen

Version 0:

Reviewer comments:

Reviewer #1

(Remarks to the Author)

The manuscript from Fernandez and colleagues with the title: "DNA2 and MSH2 activity collectively mediate chemically stabilized G4 for efficient telomere replication" is well written and built up on their previous work in which they showed that in vitro DNA2 cleaves G4 DNA. They revealed that DNA2 as well as MSH2 acts at G4s in vitro and in cells. mSH2 is essential for DNA2 activity. In DNA2 deficient cells more G4s are detected and chromatin is more accessible for G4 ligands (PIPER and TMPyP4). By single molecular assays (SMARD) they determined that DNA2 function at G4s is particularly important at telomeres during replication. They postulate a model in which MSH2 supports DNA2 G4 cleavage and by this support telomere replication. The current version of the manuscript gives a nice idea about a double mechanism on G4 resolution/excision, but it lacks mechanistic insights on how this happens.

More specific comments:

Figure 1: here they aimed to understand if DNA2 also cleaves in cells G4s. For this they saw in IF a few overlaps. This is ok, but it is not scientific to state a few. Here a quantification and statistics are needed. Also it is required to show G4 DNA2 colocalization in detail. Also why, if they focus already in Figure 1 on telomeres not do a co-staining with a telomeric probe to observe DNA2, G4 and telomeres? What is the control? Same for the IF that they used to quantify G4 in DNA2 deficient cells. 15 cells is not correct to use for quantification in 1D, also for all imaging experiments the number of biological replicates is missing. Here in sufficient detail is presented. How were these samples analysed? Overall changes or foci per cell? What is the negative control?

Further it is not clear why the author focuses on telomeres in the last part of Figure 1 and not at global DNA replication changes. Here a smooth transition is needed.

Figure 2. the authors have identified by MS a variety of DNA2 interacting proteins, why have they focused on MSH2 and is the complex DNA dependent? Also more details on the complex composition would be nice to know. The gel shift assay in 2e is in a very poor state and needs adjustment to have it in a publication quality (what is the Kd?). Again Figure 2F the quantification is missing.

Regarding the MutS α /beta discrimination, it is interesting that they show interaction with MSH2-6 which form alpha complex, but not MSH3 which is instead part of beta one.

However, here data on G4 accumulation and telomere defects in MSH2^{-/-} cells were shown. In KO MSH2, you are affecting also MutS β complex, so you can't conclude on the specificity of the consequences. A better assay would be to test the MSH6 KO

The last part of the paper connects G4 stabilization and relative effects to natural compounds.

It is interesting but it misses the point of connecting this with the delivered message from the first part of the manuscript. If this part plays an important role in the story, I would add it in the title of the manuscript.

In the last two paragraphs and in the discussion, they investigated the dual role of MutS α in G4 excision repair and FANCD1-mediated unwinding. However here no data are shown. This is too weak to make a strong conclusion out of this, also it is not reflected in the discussion (e.g. pathway choice). In order to address this point experimental data are required.

Last in the discussion, the authors have mentioned the possible activation of ALT, here it is essential to provide data

showing any increase in ALT phenotype after the same conditions.

Reviewer #2

(Remarks to the Author)

G quadruplex (G4) is a strong barrier for DNA metabolic pathways. To resolve the G4 burden, the cell relies on a set of protein factors including single-stranded DNA binding protein, helicases, and nucleases to unfold or cleave G4 DNA. Dna2 has been previously reported to cleave G4 DNA substrate. This manuscript by Fernandez et.al tested the role of Dna2-catalyzed G4 cleavage in the replication of telomeric DNA and its stimulation by the MSH2/6 complex. Moreover, by comparison to FANCI, a helicase known to unwind G4 DNA, Dna2-catalyzed G4 was found to be less sensitive to G4 stabilizing agents, which indicates a pivotal role of DNA2 in G4 processing telomere maintenance. However, this telomere phenotype of DNA2 may be also derived from the role of DNA2 in lagging strand maturation and DNA end processing. The finding of DNA2 stimulation by MSH2/6 is intriguing, it is, however, not well developed. Moreover, some of the data presented are of low quality and lack methodological details, making interpretation difficult. Overall, the quality of this research is not suitable for publication in Nature Communications. There are a few suggestions as depicted below that may strengthen this manuscript for a more specialized journal.

1. Figure 1a, quantification is necessary to strengthen the conclusion of co-localization.
2. Figure 1b, the main text mentioned Poldelta was used to reconstitute the reaction, however, the legend indicates Polbeta was used in this panel, which was very confusing. If it is Polbeta catalyzed extension, there are no experimental details. If it is Poldelta, it is not clear whether it is the catalytic subunit or the holoenzyme complex. The capability of Polbeta in the displacement of the 28-nt downstream strand is striking.
3. Figure 1e, knowing that DNA2 plays a general role in DNA replication. A control of non-telomere locus is needed to ascertain the role of DNA2 in telomeric G4 processing.
4. Figure 2e, the panel was labeled as MSH2, while the text mentioned it as MutSalpha. If MSH2 was used in this assay, the MSH2/6 complex should be used instead with the MSH2/3 complex as a control. Moreover, Is the G4 substrate used in this panel the same as 2g and 2h? and how much protein was used in the EMSA? The binding of MSH2 to the G4 substrate seems poor. The binding affinity (e.g., Kd) should be determined to strengthen the conclusion.
5. Figure 2f, again quantification is necessary to strengthen the conclusion of co-localization.
6. Figure 2g, the level of DNA2 stimulation by MutSalpha is inconsistent with the data presented in Supplementary Figure 3. Again, it is not clear how much protein was used in this assay.
7. Figure 2f, MutSalpha was labeled, but none of the reactions contains MutSalpha. If so, it should be removed.
8. Figure 4e and 4f, G4 cleavage by DNA2 is not convincing. Including MutSalpha may help strengthen these experiments.
9. Supplementary Figure 7, the formation of G4 should be tested using KCl vs. LiCl.
10. The biochemical assays were conducted in extremely low salt conditions, which doesn't represent physiological conditions. Some key experiments should be repeated with 150 mM KCl.
11. 500 ng Poldelta was used in the primer extension assay, which is a lot of protein.

Reviewer #3

(Remarks to the Author)

Fernandez and colleagues provide novel insights into the role of DNA2 in G-quadruplex (G4) resolution, particularly in cooperation with MSH2. The data are well presented, and the experiments are well-designed to address the key scientific questions. The findings are significant for understanding the mechanisms of genome stability maintenance, and the study is suitable for publication after following revisions.

1. The authors present immunofluorescence (IF) data showing endogenous DNA2 and G4 co-localization in cells. However, it would be helpful to provide a quantification of the co-localization percentage. Additionally, I am curious whether treatment with the G4 stabilizers such as TMPyP4 increases the co-localization between DNA2 and G4.
2. The authors utilize DNA2[±] MEF cells in their experiments, investigating G4 accumulation, telomere DNA replication, and abnormal telomere quantification. However, it is important to also show the DNA2 protein levels in WT and DNA2[±] MEF cells to clarify the extent of DNA2 reduction and its correlation with the observed phenotypes.
3. Based on the data, MutS α (MSH2-MSH6) is involved in both DNA2-mediated cleavage and FANCI-mediated unwinding of G4 structures. Given this, it is intriguing that MSH2^{-/-} cells exhibit a telomere replication length and abnormal telomere ratio similar to that observed in DNA2[±] cells. Could the authors provide a mechanistic explanation for this similarity? Additional discussion or data addressing whether MSH2 influences DNA2 recruitment or activity at telomeres would be beneficial.
4. The manuscript suggests that MSH2 binds directly to G4 structures and interacts with DNA2, yet DNA2 retains some G4 cleavage activity in the absence of MSH2. This raises the question of how MSH2 facilitates DNA2-mediated G4 cleavage. Specifically: Does MSH2 enhance DNA2's binding to G4 structures? Does MSH2 directly modulate DNA2 enzymatic activity? It would be useful to examine DNA2-G4 co-localization in MSH2^{-/-} cells at the cellular level. Furthermore, biochemical assays (such as EMSA or pull-down experiments) comparing DNA2 WT and catalytically inactive mutants in the presence and absence of MSH2 would provide direct evidence for whether MSH2 affects DNA2-G4 interactions.
5. In Fig. 2c-d, the DNA contribution has not been ruled out. And the interactions between MSH2, MSH6, and DNA2 could be further validated with direct pull-down assays.

Reviewer comments:

Reviewer #1

(Remarks to the Author)

We would like to thank the authors for substantial improving the manuscript and addressing most questions. Fernandez et al. described a dual role of DNA2 in resolving or cleaving DNA G4s, in cooperation with MutS α complex, to promote a successful DNA replication, especially at telomeres.

Although many queries were addressed my main concern remains: the absence of a proposed mechanism by which DNA2 G4 cleavage should be preferentially remove harmful G4 at telomeres rather than known and well-characterized helicases, like BLM or WRN, especially in "untreated conditions", where natural G4 stabilizers are not present

Here more in details some questions:

However, a mechanism underlying this dual pathway hasn't be described or discussed:

1)Figure 1D, E: G4 levels increased by inhibiting DNA2 or in DNA2 +/- genetic background: does this increase specifically happen at telomeres? A co-staining with DNA FISH may address this question. If this is not the case, how would this general increase in G4 levels fit with the following data on telomeric DNA replication issues?

2)Supp figure 4: did they test G4-containing sequences among the non-telomere fibers?

3)In figure 2E they showed that MutS α , but not MutS β , binds G4 DNA substrates; however, in Sakellariou, Despoina et al. (2022), they showed that purified MutS β binds and destabilizes G4 structures in vitro as well

4)G4 unwinding during telomere replication is mainly mediated by RecQ helicases, like WRN and BLM. According to their data on telomere replication defects, DNA2 or MSH α deficiencies are sufficient to significantly impair telomere replication. it is important to address if G4 cleavage is the preferential mechanism to remove harmful G4 at telomeres? Did the authors check the activity of known telomere helicases in DNA2- or MutS α -deficient cells?

Reviewer #2

(Remarks to the Author)

My previous concerns have been carefully addressed. The overall quality of the manuscript has been greatly improved. The newly added data largely strengthened the conclusions.

Reviewer #3

(Remarks to the Author)

The authors have addressed all my concerns in the revised version. I believe the manuscript is now suitable for publication.

Version 2:

Reviewer comments:

Reviewer #1

(Remarks to the Author)

We thank the authors for addressing our concerns and congratulations on this nice manuscript.

Responses to the reviewers' comments

Reviewer #1 (Remarks to the Author):

The manuscript from Fernandez and colleagues with the title: "DNA2 and MSH2 activity collectively mediate chemically stabilized G4 for efficient telomere replication" is well written and built up on their previous work in which they showed that in vitro DNA2 cleaves G4 DNA. They revealed that DNA2 as well as MSH2 acts at G4s in vitro and in cells. mSH2 is essential for DNA2 activity. In DNA2 deficient cells more G4s are detected, and chromatin is more accessible for G4 ligands (PIPER and TMPyP4). By single molecular assays (SMARD) they determined that DNA2 function at G4s is particularly important at telomeres during replication. They postulate a model in which MSH2 supports DNA2 G4 cleavage and by this support telomere replication. The current version of the manuscripts gives a nice idea about a double mechanism on G4 resolution/excision, but it lacks mechanistic insights on how this happens.

More specific comments:

1. Figure 1: here they aimed to understand if DNA2 also cleaves in cells G4s. For this they saw in IF a few overlaps. This is ok, but it is not scientific to state a few. Here a quantification and statistics are needed.

We have quantified co-localization of DNA2 and G4 in IF experiments. Using a software, Imaris, we have created spot maps in the nucleus and counted all co-localized foci within 100 nm of each other (page 6 lines 13-20). The quantitative data with statistics are shown in Figure 1b. We have updated the figure legend (page 25 lines 4-9).

We have also included methodological details for spot finding and co-localization analysis (page 20 lines 15 through page 21 line 2). Additionally, we emphasized in the discussion the fact that this analysis is done using super resolution microscopy, allowing us to more stringently search for co-localization because we can limit analyses to closer distances than can be performed with standard confocal microscopy (page 14, lines 3-5).

2. Also, it is required to show G4 DNA2 colocalization in detail. Also why, if they focus already in Figure 1 on telomeres, not do a co-staining with a telomeric probe to observe DNA2, G4 and telomeres?

We have added zoomed panels to Figure 1a showing co-localization of G4 and DNA2 (page 25 lines 5-8).

We have performed co-IF experiments with DNA2 and G4 and added telomere signal using FISH in Figure 3c. We have extensively quantified co-localization of DNA2 and G4 specifically at telomeres in Figure 3e (page 9, lines 19-21 and page 27 lines 16-17). We have included details on this combination of FISH and ICC protocol (page 20, lines 7-13).

3. What is the control? Same for the IF that they used to quantify G4 in DNA2 deficient cells. 15 cells are not correct to use for quantification in 1D, also for all imaging experiments the number of biological replicas is missing. Here in sufficient detail is presented. How where these samples analyzed? Overall changes or foci per cell? What is the negative control?

We have added a negative control by pre-incubating the G4 antibody with G4 containing oligos. We stained DNA2^{+/-} cells and showed the results in Supplementary Figure 1. We found that pre-incubating the G4 antibody with G4 containing oligos eliminated all G4 antibody staining from cells (page 6 lines 9-12).

Additionally, we have included information about cell counts and the number of biological replicates to the figure legends and to the materials and methods section. We have also added more detail about how cells were analyzed. Briefly, we measured the area of the foci divided by the area of the nucleus. This density measurement was compared across different conditions (page 20 lines 5-7).

Approximately 40 cells are used for quantification analysis (Figure 1b, e, Figure 2j, Figure 3b, 3d, 3e, and Figure 4h).

4. Further it is not clear why the authors focus on telomeres in the last part of Figure 1 and not at global DNA replication changes. Here a smooth transition is needed.

Since telomeres have a high frequency of guanines and are known to be enriched in G4, we measured replication rates specifically in telomeres to assess the impact of G4 on replication in cells. We have now added a short explanation to the main text for a smooth transition (Page 7, lines 15-16).

Additionally, we have analyzed replication outside of telomeres and see no significant difference in replication rates upon DNA2 knockdown (Supp. Fig 4 and page 8 lines 1-2).

5. Figure 2. the authors have identified by MS a variety of DNA2 interacting proteins, why have they focused on MSH2 and is the complex DNA dependent?

Also, more details on the complex composition would be nice to know.

The gel shift assay in 2e is in a very poor state and needs adjustment to have it in a publication quality (what is the Kd?).

Again, Figure 2F the quantification is missing.

Based on the literature, MSH2 forms a complex with MSH6 as MutS α , which recognizes mismatched DNA loops. During DNA replication and dsDNA separate into ssDNA molecules, the genome may form larger G4 loops, particularly in centromere and telomere regions. Therefore, we hypothesize that MutS α may bind to such loop structures and recruit DNA2 to resolve them. This is the rationale why we choose MSH2 as a focus for our current study (page 8 lines 18-22).

Our pull down and MS complex composition information has been uploaded to the Proteome Xchange database, in which we showed that MSH2 was the most prevalent protein found (Page 8 line 15-16 in manuscript).

In response to the reviewer's question if the MutS α and DNA2 interaction is dependent on DNA, we have also done the pull-down and western blotting with total cell extracts that are treated with benzonase, which digest both DNA and RNA. We found that the association of DNA2 and MSH2 is not dependent on DNA (The results are attached below).

Our western blots in figure 2c and 2d show that MSH2 and MSH6 (MutSa) bind to DNA2. MSH3 shows no apparent binding to DNA2, excluding interaction with MutS β . We also show that MutSa and not MutS β stimulates DNA2 activity (Figure 2l, page 9 lines 8-11). Additionally, we show below that MutSa, but not MutS β , shows strong binding to G4.

The experiment to show that MutSa binds to G4 DNA structure (EMSA) has been replaced by pull-down using G4 DNA as a bait and western blotting to detect both MutSa and MutS β components (methodological details on page 18 lines 4-18). We found that only MutSa binds to G4 DNA. The binding constant (Kd) was also determined by sigmoid plotting (Figure 2e, 2f, 2g 2h, page 9 lines 1-5). We found that the Kd of MSH2 to G4 was 48 nM and for MSH6 it was 43 nM. We included experimental details of this experiment to the methods section (page 18 lines 4-18) and updated the figure legends (page 26 lines 12-23) including specific details on protein amounts used in the reactions.

We have added quantification of the co-association of G4 and MSH2 in cells as Figure 2j (page 9 lines 4-7). The figure legend was also updated (page 26 line 25 through page 27 line 1).

6. Regarding the MutSa/alpha/beta discrimination, it is interesting that they show interaction with MSH2-6 which form alpha complex, but not MSH3 which is instead part of beta one. However, here data on G4 accumulation and telomere defects in MSH2 $-/-$ cells were shown. In KO MSH2, you are affecting also MutS β complex, so you can't conclude on the specificity of the consequences. a better assay would be to test the MSH6 KO

We have included the demanded experiments on G4 accumulation in MSH6 KO cells and show a similar increase in G4 (Figure 3 a and b, page 9 lines 13-15). We have also tested this by telomere defect accumulation and show that MSH6 KO cells have a similar increase in telomere defects (Figure 3 j and k, page 10 lines 8-11, page 27 line 24 through page 28 line 2). Thus, we can confirm that the KO of the MutSa complex, instead of MutS β complex, causes G4 accumulation and telomere defects. New panel l in figure 2 was included to show that only MutSa stimulates the DNA2 G4 removal activities, but not MutS β (page 9 lines 8-11). We have updated the figure legend accordingly (page 27 lines 3-5). We have also added these results to the discussion section (page 15, lines 5-8) showing that MutSa is responsible for these consequences.

7. The last part of the paper connects G4 stabilization and relative effects to natural compounds. It is interesting but it misses the point of connecting this with the delivered message from the first part of the manuscript. If this part plays an important role in the story, I would add it in the title of the manuscript.

We agree with the reviewer and have added the suggested key words to the manuscript title (page 1 line 1).

8. in the last two paragraphs and in the discussion, they investigated the dual role of mutsAlpha in G4 excision repair and FANCI-mediated unwinding. **However here no data are shown.** This is too weak to make a strong conclusion out of this, also it is not reflected in the discussion (e.g., pathway choice). In order to address this point experimental data are required.

FANCI-mediated G4 unwinding assay was included as a control to illustrate that the G4 stabilizers are not able to block the removal of G4 by DNA2 (Figures 4e, f) while they block the FANCI helicase activity (Figure 4c, d, page 11 lines 17-20)

We have added discussion here about the choice of helicase vs nuclease mediated G4 repair in the discussion. On page 14, lines 8-16, we discussed about how helicases are blocked by the presence of G4 stabilizers, and thus when G4 stabilizers are used we expect that nuclease mediated repair becomes critical.

9. Last in the discussion, the authors have mentioned the possible activation of ALT, here it is essential to provide data showing any increase in ALT phenotype after the same conditions.

Previous research has shown that both MutSa and DNA2 limit ALT activity. We have added citations, (Barroso-González, J. et al. Anti-recombination function of MutSa restricts telomere extension by ALT-associated homology-directed repair. Cell reports 37, 110088 (2021); and Jiang, H. et al. BLM helicase unwinds lagging strand substrates to assemble the ALT telomere damage response. Molecular Cell 84, 1684-1698.e1689 (2024), in which it has been demonstrated that ALT increases in cells lacking these proteins to the Discussion section (Page 15, lines 10-14). The data showing an increase in ALT phenotype after the same conditions were included in another manuscript which is also pending with Nature Communications currently (NCOMMS-24-74541).

Reviewer #2 (Remarks to the Author):

G quadruplex (G4) is a strong barrier for DNA metabolic pathways. To resolve the G4 burden, the cell relies on a set of protein factors including single-stranded DNA binding protein, helicases, and nucleases to unfold or cleave G4 DNA. Dna2 has been previously reported to cleave G4 DNA substrate. This manuscript by Fernandez et.al tested the role of Dna2-catalyzed G4 cleavage in the replication of telomeric DNA and its stimulation by the MSH2/6 complex. Moreover, by comparison to FANCI, a helicase known to unwind G4 DNA, Dna2-catalyzed G4 was found to be less sensitive to G4 stabilizing agents, which indicates a pivotal role of DNA2 in G4 processing telomere maintenance. However, this telomere phenotype of DNA2 may be also derived from the role of DNA2 in lagging strand maturation and DNA end processing. The finding of DNA2 stimulation by

MSH2/6 is intriguing, it is, however, not well developed. Moreover, some of the data presented are of low quality and lack methodological details, making interpretation difficult. Overall, the quality of this research is not suitable for publication in Nature Communications. There are a few suggestions as depicted below that may strengthen this manuscript for a more specialized journal.

1. Figure 1a, quantification is necessary to strengthen the conclusion of co-localization.

We have quantified co-localization of DNA2 and G4 in IF experiments. Using a software, Imaris, we have created spot maps in the nucleus and counted all co-localized foci within 100 nm of each other (page 6 lines 13-20). The quantitative data with statistics are shown in Figure 1b. We have updated the figure legend (page 25 lines 4-9).

We have also included methodological details for spot finding and co-localization analysis (page 20 lines 15 through page 21 line 2). Additionally, we emphasized in the discussion the fact that this analysis is done using super resolution microscopy, allowing us to more stringently search for co-localization because we can limit analyses to closer distances than can be performed with standard confocal microscopy (page 14, lines 3-5).

2. Figure 1b, the main text mentioned Poldelta was used to reconstitute the reaction, however, the legend indicates Polbeta was used in this panel, which was very confusing. If it is Polbeta catalyzed extension, there are no experimental details. If it is Poldelta, it is not clear whether it is the catalytic subunit or the holoenzyme complex. **The capability of Polbeta in the displacement of the 28-nt downstream strand is striking.**

The experiment in the previous Figure 1b (now Figure 1c) was designed to test the role of DNA2 nuclease activity in removal of G4 DNA. Currently, which DNA polymerase extend the gap is not defined. When we tested both Pol beta and Pol delta (Holoenzyme with all four subunits), both worked. We apologize for the mislabeling in the previous figure panel and figure legends. We have now presented the data with both Pol delta and Pol beta in the Figure 1c in replacement of previous Figure 1b and made consistent statements in the Figure 1c legends and in the text (Page 7 lines 1-6 and page 25 lines 10-14).

We have now added experimental details for both DNA Pol delta and Pol beta in the Methods section as well as in the figure legends (Page 19 lines 8-17, page 25 lines 10-14).

DNA Pol beta is known to be less processive. In our reaction, the presence of DNA2 with its helicase activity may make the Pol beta to be more processive.

3. Figure 1e, knowing that DNA2 plays a general role in DNA replication. A control of non-telomere locus is needed to ascertain the role of DNA2 in telomeric G4 processing.

We have quantified replication rates outside of telomeres and found no significant difference in replication rates with or without DNA2 (Supp. Fig 4 and page page 8 lines 1-2). Data presented in Figure 3c-e also indicate that MSH2 facilitates DNA2 colocalization to G4 in the telomere regions preferentially (page 9 lines 19-22). This is consistent with our previous observation that DNA2

preferentially binds to difficult-to-replicate (DTR) regions such as centromere (Li et al., 2018, EMBO J.).

4. Figure 2e, the panel was labeled as MSH2, while the text mentioned it as MutS alpha. If MSH2 was used in this assay, the MSH2/6 complex should be used instead with the MSH2/3 complex as a control. Moreover, Is the G4 substrate used in this panel the same as 2g and 2h? and how much protein was used in the EMSA? The binding of MSH2 to the G4 substrate seems poor. The binding affinity (e.g., Kd) should be determined to strengthen the conclusion.

The experiment to show that MutSa binds to G4 DNA structure (EMSA) has been replaced by pull-down using G4 DNA as a bait and western blotting to detect both MutSa and MutSβ components. We found that only MutSa binds to G4 DNA. The binding constant (Kd) was also determined by sigmoid plotting (Figure 2e, 2f, 2g 2h, page 9 lines 1-5). We found that the Kd of MSH2 to G4 was 48 nM and for MSH6 it was 43 nM. We included experimental details of this experiment to the methods section (page 18 lines 4-18) and updated the figure legends (page 26 lines 12-23) including specific details on protein amounts added.

MutSa complex instead of MSH2 was used in our previous and current experiments. The G4 substrate used in Figure 2e is the same as the one in Figure 2g (now figure 2k). The previous figure 2h has been removed as it is redundant with Figures 4e and 4f.

The amount of the proteins used in the experiments is indicated in Figure 2 legends (page 26 lines 12-23 and page 27 lines 1-5).

5. Figure 2f, again quantification is necessary to strengthen the conclusion of co-localization.

We have added quantification of the co-association of G4 and MSH2 as Figure 2j (page 26 lines 25-26) and added this to the results section (page 9 lines 5-7).

6. Figure 2g, the level of DNA2 stimulation by MutS alpha is inconsistent with the data presented in Supplementary Figure 3. Again, it is not clear how much protein was used in this assay.

We have now compared the stimulation levels by MutSa and MutSβ side by side in a new reaction. The results are presented in Figure 2l, indicating that MutSa greatly stimulates the DNA2 mediated cleavage activity, while the MutSβ does not (page 9 lines 8-11). This experiment was done using a radiolabeled G4 substrates and the amounts of DNA2, MutSa, MutSβ and XPF proteins in the reactions are indicated in the figure legend (page 27 lines 1-5).

We have also added more thorough experimental details (page 18 line 20 through page 19 line 6).

7. Figure 2f, MutSalpha was labeled, but none of the reactions contains MutSalpha. If so, it should be removed.

In the original Figure 2f, MSH2 was labeled, which was correct as MSH2 antibody was used to demonstrate the co-localization of MSH2 and G4 foci in the immuno-florescence (IF) staining experiments (now Figure 2j and i).

However, this may have been in reference to Figure 2h. This figure has been replaced by figures 4e and 4f (as suggested in point 8 below) where DNA2 is incubated with MutS α as indicated in the response to question 8. This new data is redundant with the original figure 2h, where we show that DNA2 nuclease activity is uninhibited by the presence of G4 stabilizers.

8. Figure 4e and 4f, G4 cleavage by DNA2 is not convincing. Including MutS α may help strengthen these experiments.

We have redone the assays by including MutS α for its stimulating effects on the DNA2 nuclease activity to show the DNA2 is able to remove the chemically stabilized G4 structures with the increased chemical stabilizer (PIPER or capreomycin) dosages (Fig. 4e, 4f, page 11 lines 18-20, page 28 lines 12-14).

9. Supplementary Figure 7, the formation of G4 should be tested using KCl vs. LiCl.

As suggested, the formation of G4 has been tested using KCl vs. LiCl. The results were presented in Supplementary Figure 2 (page 6 lines 24-25).

10. The biochemical assays were conducted in extremely low salt conditions, which doesn't represent physiological conditions. Some key experiments should be repeated with 150 mM KCl.

We acknowledge that for many DNA polymerase and nuclease biochemical assays, KCl or NaCl concentrations are used in the range of 50-300 mM, with 150 mM often being considered a physiological concentration. However, the capacity of DNA polymerase delta in resolution of G4 structures is sensitive to the concentration of KCl. It was stalled at concentration of 20-40 mM (Supplementary Figure 7b). To see if nucleases or helicases help Pol delta process through the G4 structures, we chose to use a 20 mM concentration. The DNA nuclease activity is more tolerant to the salt concentration. We have used 100 mM of KCl in our G4 cleavage biochemical assays (Fig 4 e and f) and indicated this in the methods section (page 19 lines 1-3).

11. 500 ng Poldelta was used in the primer extension assay, which is a lot of protein.

We have redone the biochemical assays showing polymerase δ extension activities in the presence or absence of G4 stabilizers with 50 ng of Pol delta. The new data now included in Figure 4 and Supplementary Figure 8. We have reflected this in the method description of the polymerase extension assay (page 22 lines 7-21).

Reviewer #3 (Remarks to the Author):

Fernandez and colleagues provide novel insights into the role of DNA2 in G-quadruplex (G4)

resolution, particularly in cooperation with MSH2. The data are well presented, and the experiments are well-designed to address the key scientific questions. The findings are significant for understanding the mechanisms of genome stability maintenance, and the study is suitable for publication after following revisions.

1. The authors present immunofluorescence (IF) data showing endogenous DNA2 and G4 co-localization in cells. However, it would be helpful to provide a quantification of the co-localization percentage. Additionally, I am curious whether treatment with the G4 stabilizers such as TMPyP4 increases the co-localization between DNA2 and G4.

We have quantified co-localization of DNA2 and G4 in IF experiments. Using a software, Imaris, we have created spot maps in the nucleus and counted all co-localized foci within 100 nm of each other (page 6 lines 13-20). The quantitative data with statistics are shown in Figure 1b. We have updated the figure legend (page 25 lines 4-9).

We have also included methodological details for spot finding and co-localization analysis (page 20 lines 15 through page 21 line 2). Additionally, we emphasized in the discussion the fact that this analysis is done using super resolution microscopy, allowing us to more stringently search for co-localization because we can limit analyses to closer distances than can be performed with standard confocal microscopy (page 14, lines 3-5).

We also compared co-localization in the presence of G4 stabilizer in supplementary Fig. 9 that shows an increase in co-localization between DNA2 and G4 (page 11 line 26 through page 12 line 4).

2. The authors utilize DNA2^{+/-} MEF cells in their experiments, investigating G4 accumulation, telomere DNA replication, and abnormal telomere quantification. However, it is important to also show the DNA2 protein levels in WT and DNA2^{+/-} MEF cells to clarify the extent of DNA2 reduction and its correlation with the observed phenotypes.

We used the same WT and DNA2^{+/-} MEF cells as we published in 2013 EMBO J. in which we showed significantly decreased expression of DNA2 in the mutant cells (see below).

3. Based on the data, MutS α (MSH2-MSH6) is involved in both DNA2-mediated cleavage and FANCI-mediated unwinding of G4 structures. Given this, it is intriguing that MSH2^{-/-} cells exhibit a telomere replication length and abnormal telomere ratio similar to that observed in DNA2^{+/-} cells. Could the authors provide a mechanistic explanation for this similarity? Additional discussion or data addressing whether MSH2 influences DNA2 recruitment or activity at telomeres would be beneficial.

In our discussion section, we have added the following paragraph to provide an explanation for the above observations:

In our analysis of the recruitment of DNA2 to G4 in telomere regions via IF and pulldown/western blotting, we found that absence of MSH2 significantly reduces DNA2 recruitment to G4 structures. MutS α facilitates the recruitment of DNA2 onto the G4 structures. Similarly, in DNA2 \pm cells, there is insufficient DNA2 protein to be recruited onto the G4 structures. Therefore, absence of MSH2 or haploinsufficiency of DNA2 may cause similar cellular phenotypes of shortened telomere length and abnormal telomere frequency (page 15 lines 22 through page 16 line 2).

In addition, we have done co-localization analyses between DNA2 and G4 in presence and absence of MSH2. We found that DNA2 and G4 shows a decrease in co-localization with telomeres in absence of MSH2 (Figure 3e, page 9 lines 19-22), suggesting that MSH2 is especially important for DNA2 recruitment to telomeric G4s. We have added this to the figure legend (page 27 lines 16-17).

4. The manuscript suggests that MSH2 binds directly to G4 structures and interacts with DNA2, yet DNA2 retains some G4 cleavage activity in the absence of MSH2. This raises the question of how MSH2 facilitates DNA2-mediated G4 cleavage. Specifically: Does MSH2 enhance DNA2's binding to G4 structures? Does MSH2 directly modulate DNA2 enzymatic activity? It would be useful to examine DNA2-G4 co-localization in MSH2 \pm cells at the cellular level.

Furthermore, biochemical assays (such as EMSA or pull-down experiments) comparing DNA2 WT and catalytically inactive mutants in the presence and absence of MSH2 would provide direct evidence for whether MSH2 affects DNA2-G4 interactions.

Our amended biochemical experiments demonstrated that MutS α , but not MutS β , significantly stimulates the DNA2's nuclease activity towards G4 substrates (Figure 2k and 2l, page 9 lines 7-10).

In addition, we have done co-localization analyses between DNA2 and G4 in presence and absence of MSH2. We found that DNA2 and G4 shows a slight decrease in co-localization in MSH2 \pm cells (significantly more pronounced in telomeres) in Figure 3c, 3d, and 3e, page 9 lines 15-22 and page 27 lines 13-18). We also show that MSH2 and MSH6 stimulates binding of DNA2 to G4 DNA (Figure 3f, page 9 lines 22-23 and page 27 lines 18-19) Thus, it is likely that MSH2 is also involved in enhancing DNA2 binding to G4 structures.

5. In Fig. 2c-d, the DNA contribution has not been ruled out. And the interactions between MSH2, MSH6, and DNA2 could be further validated with direct pull-down assays.

We have tested this interaction with benzonase treatment and found that the association of DNA2 and MSH2 remains even without DNA. Results were shown in the response to reviewer 1, question 5 above.

Responses to the reviewers' comments

We would like to thank the authors for substantially improving the manuscript and addressing most questions. Fernandez et al. described a dual role of DNA2 in resolving or cleaving DNA G4s, in cooperation with MutS α complex, to promote a successful DNA replication, especially at telomeres.

Although many queries were addressed my main concern remains: the absence of a proposed mechanism by which DNA2 G4 cleavage should be preferentially remove harmful G4 at telomeres rather than known and well-characterized helicases, like BLM or WRN, especially in "untreated conditions", where natural G4 stabilizers are not present

Here more in details some questions:

However, a mechanism underlying this dual pathway hasn't be described or discussed:

1) Figure 1D, E: G4 levels increased by inhibiting DNA2 or in DNA2 +/- genetic background: does this increase specifically happen at telomeres? A co-staining with DNA FISH may address this question. If this is not the case, how would this general increase in G4 levels fit with the following data on telomeric DNA replication issues?

We stained for G4 at telomeres using a combination of G4 staining and telomere FISH. We found that there was a significant increase of G4 at telomeres (Supplementary Figure 3, page 7 lines 11-17). This gives further evidence that the increase of G4 specifically at telomeres is responsible for the telomere replication defect and telomere dysfunction.

2) Supp figure 4: did they test G4-containing sequences among the non-telomere fibers?

G4-containing sequences are heterogeneous throughout the genome. They are mostly found in repetitive guanine containing regions such as the centromere and telomere. We note that SMARD experiments are mostly performed using telomere and centromere specific probes though the assay can be performed if a specific probe is identified. In the current study, we use telomeres as a model to explore the interrelationship among the G4 enrichment, DNA replication dynamics, telomere abnormality, and the role of DNA2. We did not foresee a strong rationale to test other various G-enriched regions.

3) In figure 2E they showed that MutS α , but not MutS β , binds G4 DNA substrates; however, in Sakellariou, Despoina et al. (2022), they showed that purified MutS β binds and destabilizes G4 structures in vitro as well

We do not preclude the possibility of MutS β binding to G4. Our work indicates that MutS α binding to G4 is more favorable than MutS β . Further, we note that the G4 oligo used in Sakellariou *et al.* is single stranded. Our *in vitro* assays used double stranded G4, which may preclude MutS β recognition.

4) G4 unwinding during telomere replication is mainly mediated by RecQ helicases, like WRN and BLM. According to their data on telomere replication defects, DNA2 or MSHalpha deficiencies are sufficient to significantly impair telomere replication. It is important to address if G4 cleavage is the preferential mechanism to remove harmful G4 at telomeres? Did the authors check the activity of known telomere helicases in DNA2- or MutSalpha-deficient cells?

To test the activity of helicases without DNA2 resolution, we treated cells with both the WRN inhibitor HRO761 and DNA2 inhibitor C5. We found that while both were able to elicit an increase in G4 at telomeres, the combination leads to an even greater increase of G4 at telomeres. This suggests that helicases are still active even when cells are deficient in DNA2 activity (Supplementary Figure 3, page 7 lines 11-17). Thus, helicase-mediated G4 resolution and nuclease-mediated removal of stabilized G4 are separate mechanisms, both of which are important for maintenance of telomere integrity.